# CONFORMALIZED INTERACTIVE IMITATION LEARNING: HANDLING EXPERT SHIFT & INTERMITTENT FEEDBACK

**Michelle Zhao[a], Reid Simmons[a], Henny Admoni[a], Aaditya Ramdas[†b], Andrea Bajcsy[†a]**
[a] Robotics Institute, School of Computer Science, Carnegie Mellon University
[b] Departments of Statistics and Machine Learning, Carnegie Mellon University
{mzhao2, hadmoni, rsimmons, aramdas, abajcsy}@andrew.cmu.edu

## ABSTRACT

In interactive imitation learning (IL), uncertainty quantification offers a way for the learner (i.e. robot) to contend with distribution shifts encountered during deployment by actively seeking additional feedback from an expert (i.e. human) online. Prior works use mechanisms like ensemble disagreement or Monte Carlo dropout to quantify when black-box IL policies are uncertain; however, these approaches can lead to overconfident estimates when faced with deployment-time distribution shifts. Instead, we contend that we need uncertainty quantification algorithms that can leverage the expert human feedback received *during deployment time* to adapt the robot's uncertainty *online*. To tackle this, we draw upon online conformal prediction, a distribution-free method for constructing prediction intervals online given a stream of ground-truth labels. Human labels, however, are intermittent in the interactive IL setting. Thus, from the conformal prediction side, we introduce a novel uncertainty quantification algorithm called intermittent quantile tracking (IQT) that leverages a probabilistic model of intermittent labels, maintains asymptotic coverage guarantees, and empirically achieves desired coverage levels. From the interactive IL side, we develop ConformalDAgger, a new approach wherein the robot uses prediction intervals calibrated by IQT as a reliable measure of deployment-time uncertainty to actively query for more expert feedback. We compare ConformalDAgger to prior uncertainty-aware DAgger methods in scenarios where the distribution shift is (and isn't) present because of changes in the expert's policy. We find that in simulated and hardware deployments on a 7DOF robotic manipulator, ConformalDAgger detects high uncertainty when the expert shifts and increases the number of interventions compared to baselines, allowing the robot to more quickly learn the new behavior. Project page at cmu-intentlab.github.io/conformalized-interactive-il/.

## 1 INTRODUCTION

End-to-end robot policies trained via imitation learning (IL) have proven to be an extremely powerful way to learn complex robot behaviors from expert human demonstrations (Schaal, 1996; Price & Boutilier, 2003; Argall et al., 2009; Levine et al., 2016; Jang et al., 2022; Chi et al., 2023; Kim et al., 2024). At the same time, distribution shift is a core challenge in this domain, hampering the reliability of deploying such robot policies in the real world (Chang et al., 2021).

One way to combat this is via uncertainty quantification. By training an ensemble of policies (Menda et al., 2019) or via monte-carlo dropout during training (Cui et al., 2019), the robot learner can detect uncertain states and *actively elicit* additional action labels from the human expert online via an interactive IL framework (such DAgger (Ross et al., 2011)). At their core, these prior uncertainty-aware IL methods look to the training demonstration data as a proxy for deployment-time uncertainty. Any human expert labels requested at deployment time using this uncertainty estimate are simply stored for later re-training; the uncertainty estimate itself is not adapted online to the expert data nor does it inform any subsequent queries during the deployment episode.

Instead, we contend that the human feedback requested and received *during* deployment time is a valuable uncertainty quantification signal that should be leveraged to *update* the robot's uncertainty estimate *online*. If properly accounted for, the updated uncertainty estimate will influence when the robot asks for more help, enabling it to targetedly probe the human expert to improve policy

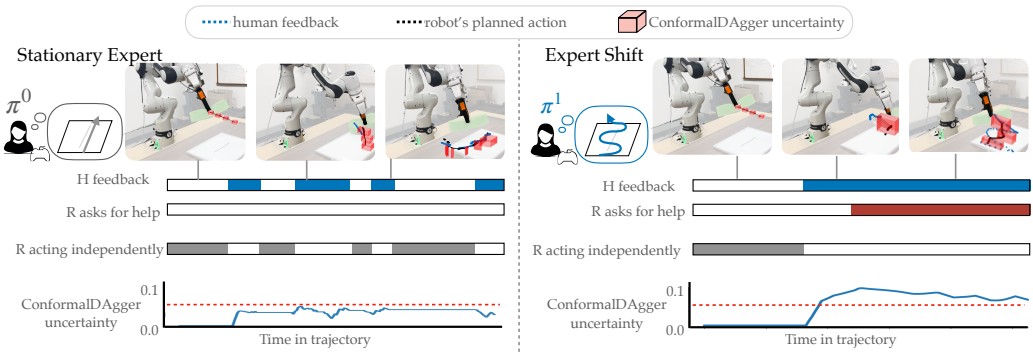

Figure 1: **Conformalized Interactive Imitation Learning.** The robot learns an initial policy via imitation learning to wipe a line drawing by following a straight path. When the initial policy is deployed, ConformalDAgger calibrates the robot's uncertainty (represented by red prediction interval boxes) based on feedback received from the human during the interactive IL loop. (left) The robot refrains from asking questions when its uncertainty is low because the expert policy at deployment time is aligned to the demonstrations on which the learner was trained. (right) When the human shifts in their task strategy, uncertainty increases and the robot starts to query for more expert labels.

performance. The challenge is how to do this online uncertainty update in the presence of the end-to-end "black box" policies underlying modern imitation learning.

To tackle this, we take inspiration from online conformal prediction (Gibbs & Candes, 2021) which is a distribution-free way to represent uncertainty via prediction sets constructed on the output of a black-box model. Our first contribution is extending online conformal prediction to the case where labels are observed intermittently, as is the case in interactive IL with an expert. Specifically, we instantiate Intermittent Quantile Tracking (IQT), an algorithm which adjusts prediction intervals online to ensure that the true label lies within the predicted interval with high-probability despite probabilistic access to labels. On standard conformal time series datasets, we empirically find that IQT achieves empirical coverage close to the desired level by boosting the size of the calibrated intervals based on the likelihood of observing feedback.

With our intermittent conformal algorithm in hand, we develop ConformalDAgger, a new interactive IL approach wherein the robot uses prediction intervals calibrated by IQT as a reliable measure of deployment-time uncertainty to actively query for more expert feedback. We instantiate ConformalDAgger in a simulated 4D robot goal-reaching task and in hardware on a 7 degree-of-freedom robotic manipulator that uses a state-of-the-art Diffusion Policy (Chi et al., 2023) learned via IL to perform a sponging task (Figure 1). We study how ConformalDAgger compares to prior uncertainty-aware DAgger methods when the deployment-time and training-time datasets are from the same distribution (left, Figure 1) as well as a shifted one (right, Figure 1). Specifically, we instantiate a potential source of distribution shift as *expert policy shift*: the training-time expert wipes a line-drawing with a straight-line path while the deployment-time expert refines their strategy to be a zigzag path (Figure 1). We find that ConformalDAgger automatically increases uncertainty online when the expert shifts, resulting in more expert labels queries compared to EnsembleDAgger and allowing our approach to rapidly learn a policy aligned with the expert's intentions.

## 2 RELATED WORK

**Interactive Imitation Learning (IL) with Online Experts.** Interactive IL is a branch of imitation learning wherein a robot learner can query a (human) expert to receive additional labels either during or after task execution (Celemin et al., 2022). The utility of in-the-loop expert interventions is mitigating distribution shift between training and deployment of the policy. Distribution shift can stem from a multitude of reasons, including covariate shift (Spencer et al., 2021) (e.g., encountering new observations of the environment during policy execution), or expert shift: wherein the expert's latent strategy changes over time (Hong et al., 2024; Sagheb et al., 2023; Xie et al., 2021) A foundational approach for interactive IL is DAgger (Dataset Aggregation) (Ross et al., 2011), which iteratively augments the training dataset by aggregating data from the expert and learner policies and assuming the expert is stationary. In the online case, the robot learner can cede control to the expert at any time to get additional state-action data (also known as *robot-gated* feedback) or the expert can actively

intervene at any time (also known as *human-gated* feedback) (Kelly et al., 2019). From the learner's perspective, a key question is when to request feedback from the expert so that it minimizes expert effort but also minimizes negative events caused by an erroneous policy (e.g., running into a wall). On one hand, prior works focus on minimizing human effort by constraining robot requests via a limited human attention model (Hoque et al., 2023) or a budget of human interventions (Hoque et al., 2021a;b). On the other hand, prior works prioritize deployment-time safety by classifying safe versus unsafe states (SafeDAgger (Zhang & Cho, 2017), Replay Estimation (Swamy et al., 2022)), estimating uncertainty via ensemble disagreement (EnsembleDAgger (Menda et al., 2019)), or Monte Carlo dropout (Cui et al., 2019). We present ConformalDAgger, which uses our novel uncertainty quantification method grounded in conformal prediction to adaptively increase the learner requests for help under uncertainty and decrease the number of requests when confident.

**Online Conformal Prediction.** Conformal prediction is a distribution-free uncertainty quantification method for constructing prediction intervals for both classification and regression problems (Angelopoulos & Bates, 2023; Romano et al., 2019; 2020; Zaffran et al., 2022), as well as for offline or online data. We focus on the *online* setting (e.g., timeseries) where uncertainty quantification is performed on streaming pairs of input-label data that are not necessarily i.i.d. (Gibbs & Candes, 2021). Broadly speaking, there are two predominant algorithms in this setting: adaptive conformal inference (ACI) (Gibbs & Candes, 2021; Gibbs & Candès, 2024; Bhatnagar et al., 2023; Zaffran et al., 2022) and quantile tracking (QT) (Angelopoulos et al., 2023). Both are online gradient descent-based methods which guarantee asymptotic coverage in the online setting. Unlike ACI which is prone to infinitely sized intervals after a series of miscoverage events, QT directly estimates the value of the empirical quantile itself, ensuring coverage with finite intervals. In this work, we relax the assumption that labels must be observed at each time point in the streaming data and extend the online conformal paradigm to ensure coverage in the intermittent label regime.

## 3 ONLINE CONFORMAL PREDICTION WITH INTERMITTENT LABELS

From the uncertainty quantification side, our core technical contribution is extending online conformal prediction to settings where ground truth labels are intermittently observed. We present our algorithm in the context of quantile tracking for relevance to our interactive IL experiments. However, we also derive an extension of ACI (Gibbs & Candes, 2021) to intermittent labels in the Appendix Section D.

**Setting.** We focus on online conformal prediction in the adversarial setting, such as time-series forecasting. This considers an *arbitrary* sequence of data points $(x_t, y_t) \in \mathcal{X} \times \mathcal{Y}$, for $t = 1, 2, ...$, that are not necessarily I.I.D. Our goal is to produce prediction sets on the output of any base prediction model such that the sets contain the true label with a specified coverage probability. Mathematically, at each time $t$, we observe $x_t$ and seek to cover the true label $y_t$ with a set $C_t(x_t)$, which depends on a base prediction model, $\hat{f} : \mathcal{X} \rightarrow \mathcal{Y}$. The base model takes as input the current $x_t$ and outputs prediction $\hat{y}_t$; in the *non*-intermittent case, we observe the ground-truth label $y_t$ after each prediction.

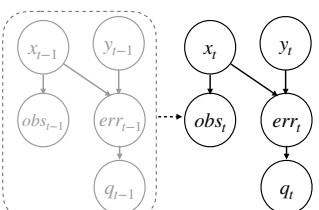

Figure 2: **Graphical Model of IQT.** IQT introduces random variable $\text{obs}_t$, which represents the receiving of ground truth observations at $t$. $\text{obs}_t$ is distributed according to $p_t$, which may depend on $x_t$ or history through $t - 1$.

**Background: Quantile Tracking.** The quantile tracking (QT) algorithm from Angelopoulos et al. (2024) implicitly seeks to track the value of the $1 - \alpha$ quantile via online gradient descent on the quantile loss (Koenker & Bassett Jr, 1978). As in Angelopoulos et al. (2024), we leverage a bounded nonconformity score function: $s : \mathcal{X} \times \mathcal{Y} \rightarrow [0, B]$ where $0 < B < \infty$, to quantify the error made by the initial prediction of the base model. We assume the nonconformity score function $s(x_t, y_t)$ is negatively oriented (lower values indicate less nonconformity or greater prediction accuracy). Let $q_t$ represents the estimated $1 - \alpha$ quantile of the score sequence $s_t, t \in \mathbb{N}$. Prediction intervals are constructed using the nonconformity score function:

$$C_t(x_t) = \{y \in \mathcal{Y} : s(x_t, y_t) \leq q_t\} \tag{1}$$

To expand or contract the prediction intervals, the level $q_t$ is adjusted via the online update:

$$q_{t+1} = q_t + \gamma_t(\text{err}_t - \alpha), \tag{2}$$

where $\gamma_t > 0$ is a time-varying step size and $\text{err}_t = \mathbb{1}_{y_t \notin C_t(x_t)}$ is the empirical miscoverage at $t$.

Intuitively, this update increases the quantile threshold when the model continuously miscovers and decreases it when the model coverage is performant. For arbitrary step size $\gamma_t$ with no assumptions on the sequence of data points $(x_1, y_1), (x_2, y_2), ...$ and an initial quantile threshold $q_1 \in [0, B]$, the quantile tracking update satisfies Equation 3 (Theorem 2 of Angelopoulos et al. (2024)):

$$\left| \frac{1}{T} \sum_{t=1}^{T} \mathbb{1}_{y_t \in C_t(x_t)} - (1 - \alpha) \right| \leq \frac{B + \max_{1 \leq t \leq T} \gamma_t}{T} \cdot ||\Delta_{1:T}||_1 \tag{3}$$

where $\Delta$ is defined $\Delta_1 = \gamma_1^{-1}$ and $\Delta_t = \gamma_t^{-1} - \gamma_{t-1}^{-1}$ for $t \geq 2$, and $\Delta_{1:T} = (\Delta_1, \ldots, \Delta_T)$. We can see that $\lim_{T \to \infty} \frac{1}{T} \sum_{t=1}^{T} 1 - \text{err}_t$ approaches $1 - \alpha$. This guarantees quantile tracking gives the $1 - \alpha$ long-term empirical coverage frequency.

**Intermittent Quantile Tracking (IQT).** Our paradigm lifts the assumption that the ground truth label $y_t$ is observed constantly. Instead, it is observed with some probability at each timestep. Let the binary random variable $\text{obs}_t \in \{0, 1\}$ represent whether the robot observes label $y_t$ at timestep $t$:

$$\text{obs}_t := \begin{cases} 1, & \text{if } y_t \text{ observed} \\ 0, & \text{otherwise} . \end{cases} \tag{4}$$

We introduce a probabilistic observation model, $p_t := \mathbb{P}(\text{obs}_t = 1 \mid x_t)$, where $p_t$ represents the probability of receiving feedback, which can be dependent on input $x_t$ (and past history, this possibility is demonstrated by the dotted line in Figure 2). A key observation under the paradigm of intermittent labels is that $\text{err}_t$ may not at every timestep be accessible to the algorithm if $y_t$ is not provided, but the value of $\text{err}_t$ exists at every timestep even if unobservable.

Our quantile tracking update under probabilistic observations, which we call Intermittent Quantile Tracking (IQT), is defined in Equation 5 as:

$$q_{t+1} = q_t + \frac{\gamma_t}{p_t}(\text{err}_t - \alpha)\text{obs}_t \tag{5}$$

**Proposition 1.** *Let $(x_1, y_1), (x_2, y_2), ...$ be an arbitrary sequence of data points, and let $s : \mathcal{X} \times \mathcal{Y} \to [0, B]$. Let $\gamma_t$ be an arbitrary positive sequence, and fix an initial threshold $q_1 \in [0, B]$. Then Intermittent Quantile Tracking (IQT) satisfies for all $T \geq 1$:*

$$\left| \frac{1}{T} \sum_{t=1}^{T} err_t - \alpha \right| \leq \frac{B + \max_{1 \leq t \leq T} \frac{\gamma_t}{p_t}}{T} \cdot ||\Delta_{1:T}||_1 \tag{6}$$

This guarantees IQT gives the $1 - \alpha$ long-term empirical coverage frequency, even with intermittent ground truth access, regardless of the underlying data generation process. While these asymptotic coverage guarantees are sound in theory, we believe it is important to acknowledge that typical robot deployment conditions are finite-horizon. Nevertheless our empirical findings in Section 5 indicate the practical utility of IQT's uncertainty estimates to obtain extra human feedback during deployment. Refer to full proof in Appendix Section C. Note that when feedback is constant (i.e. $p_t = 1$), $\text{obs}_t = 1$ for all timesteps, and IQT reduces to online quantile tracking with arbitrary step sizes.

*Practical Note: Choosing Gamma.* In practice, an important decision is the choice of $\gamma_t$. Prior works in quantile tracking with constant labels (Angelopoulos et al., 2023) choose $\gamma_t = \text{lr}\hat{B}_t$, where $\hat{B}_t := \max_{t-k \leq r \leq t-1} s(x_r, y_r)$, $k$ represents a look-back window across the previous timesteps with observed ground truth labels, and lr is a small constant. On our domains, we find that choosing $\gamma_t$ such that it contains $\hat{B}_t$ improves empirical coverage over the choice of a constant $\gamma_t = \gamma$.

### 3.1 EXPERIMENTS: INTERMITTENT QUANTILE TRACKING ON STANDARD TIMESERIES DATA

We first empirically evaluate our approach for intermittent conformal quantile tracking on time series benchmarks used in the online conformal prediction literature. The goal of testing on standard conformal datasets is to evaluate (1) how different choices of step size $\gamma_t$ affect the empirical coverage of IQT, and (2) validate that under probabilistically intermittent observations, IQT maintains coverage close to the desired level. We use the findings from these experiments to inform how we leverage intermittent quantile tracking in the interactive imitation learning domain.

**Setup.** We test on three benchmark datasets from Angelopoulos et al. (2023): (1) Amazon stock prices, (2) Google stock prices (Nguyen, 2018), and the (3) Elec2 dataset (Harries, 1999). We test

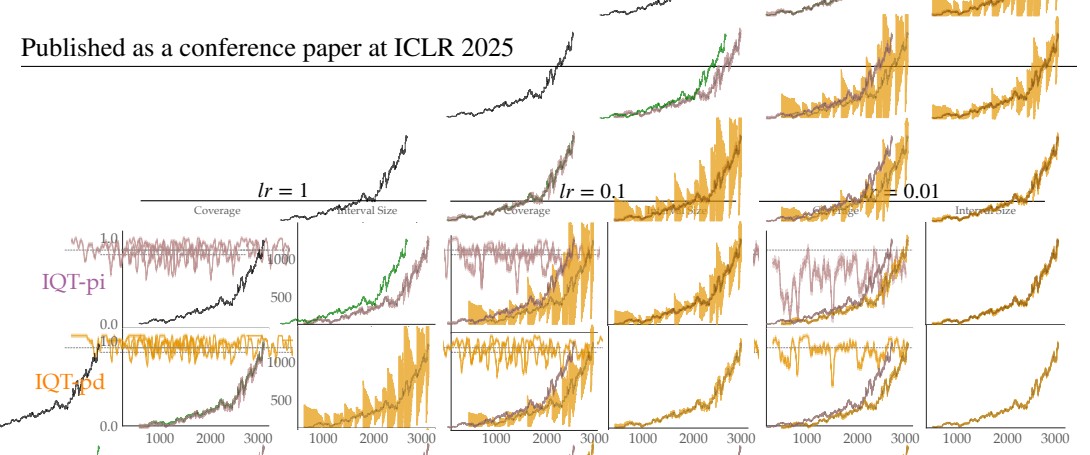

Figure 3: **IQT on Amazon Stocks with AR base model: Coverage & Interval Visualization.** We set $p_t = 0.1, \forall t$ to see true price labels only 10% of the time and show the prediction interval for 1 seed and coverage averaged over 5 seeds (shaded error on coverage plots show std deviation). **IQT-pd** is prone to large intervals when lr is large, and boosts coverage over **IQT-pi** when lr is small.

four base prediction models, $\hat{f}$, all trained via darts (Herzen et al., 2022) to see if IQT consistently maintains coverage close to the desired level. We present the Autoregressive (**AR**) model with 3 lags for brevity in the main text and defer the other model results to the Appendix. Our nonconformity score is the asymmetric (signed) residual score. We measure (1) marginal coverage over the time series, (2) longest miscoverage error sequence, and (3) mean prediction interval size.

**Observation Models.** We test three different observation frequencies: (1) *infrequent* $p_t = 0.1, \forall t$, (2) *partial* $p_t = 0.5, \forall t$, and (3) *frequent* $p_t = 0.9, \forall t$. We focus in this section on the results for *infrequent* observations, and discuss further results in the Appendix Section E.

**Methods.** We compare two variants of our IQT algorithm by controlling if time-varying step-size $\gamma_t$ in Equation 5 depends or does not depend on $p_t$. For each variant, we test lr $\in [1, 0.1, 0.01]$. **IQT with a $p_t$-independent update** (**IQT-pi**) uses $\gamma_t = \text{lr}\hat{B}_t p_t$. The quantile tracking update reduces to $q_{t+1} = q_t + \text{lr}\hat{B}_t(\text{err}_t - \alpha)\text{obs}_t$. Since the $p_t$ term cancels out in the update, we refer to this model as the p-*independent* update. **IQT with a $p_t$-dependent update** (**IQT-pd**) uses $\gamma_t = \text{lr}\hat{B}_t$. Since the $p_t$ term remains in the update $q_{t+1} = q_t + \frac{\text{lr}\hat{B}_t}{p_t}(\text{err}_t - \alpha)\text{obs}_t$, we refer to this model as the p-dependent update. For both variants, we set our desired coverage to $1 - \alpha = 0.9$ for all experiments. We set lookback window $k = 100$ timesteps for the Amazon stock price data.

**Results.** We center our discussion on the Amazon stock price results for **IQT-pd** and **IQT-pi** with AR base model under ground truth labels observed with $p_t = 0.1$ frequency, as the intermittent setting is our focus. See Appendix Section E for further results on other base models and datasets. Inversely scaling the quantile tracking step size by $p_t$ causes **IQT-pd** to construct larger intervals than **IQT-pi**. Figure 3 shows the prediction coverage as a moving average (window=50) for 5 random seeds and interval sizes for one seed. When lr is high (1.0), **IQT-pi** and **IQT-pd** achieve comparable coverage, but **IQT-pd** is prone to constructing much larger prediction intervals than **IQT-pi**. Smaller and midsized learning rates (lr = 0.01, 0.1) regulate the size of **IQT-pd** intervals, leading to tight prediction intervals which maintain desired coverage levels.

## 4 ConformalDAgger: A Calibrated Approach to Asking for Expert Feedback

Since intermittent quantile tracking enables us to rigorously quantify the learner's uncertainty despite the fact that the labels are only revealed intermittently (i.e., when the expert intervenes), we can develop ConformalDAgger: a new way for the robot to tradeoff between acting autonomously and strategically asking for help when uncertainty increases.

**Setup.** ConformalDAgger treats the robot's policy as the base model $\hat{f} := \pi^r$ on which we perform intermittent quantile tracking. The initial novice policy $\pi_0^r : \mathcal{X} \to \mathcal{A}$ is trained on initial demonstration data $\mathcal{D}_0$ of task $\mathcal{T}$ performed by the expert $\pi_0^h$. Here, $\mathcal{X}$ represents the policy's inputs (e.g., image observations, proprioception), $\mathcal{Y} \equiv \mathcal{A}$ are the labels representing the robot's actions (e.g., future end-effector positions). During deployment, the robot policy generates a sequence of input-predicted action pairs $(x_t, a_t^r) \in \mathcal{X} \times \mathcal{Y}$ for $t = 1, 2, ...,$ that are temporally correlated. Relatedly, for each input $x_t$ the learner observes, there is a corresponding expert action $(x_t, a_t^h) \in \mathcal{X} \times \mathcal{Y}$ that is the ground-truth label we seek to cover via our IQT prediction intervals $C_t(x_t)$.

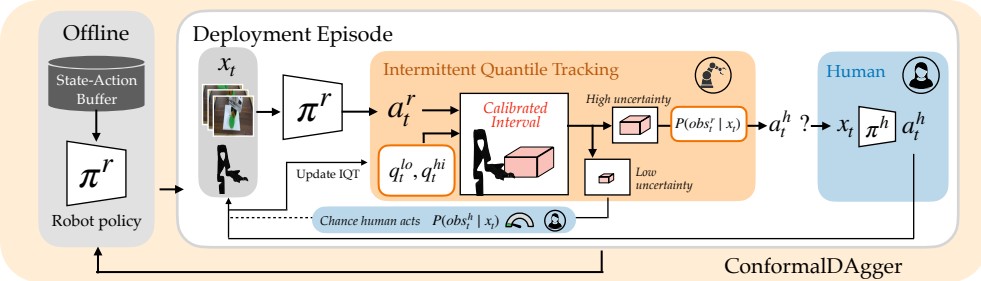

Figure 4: **ConformalDAgger Framework.** After obtaining an initial learner policy $\pi^r$ (left), ConformalDAgger calibrates uncertainty *during* the interactive deployment episode with the expert via intermittent quantile tracking (center). When the size of the uncertainty intervals is high, the robot actively queries the user for feedback. When uncertainty is low, the robot executes its predicted action and the human may independently intervene with some low probability. After deployment episode ends, the data is aggregated and the learner retrained (arrow from right to left).

---

**Algorithm 1** ConformalDAgger *(changes from DAgger ([Ross et al., 2011](#)) highlighted)*

1: Collect initial demonstration data $\mathcal{D}_0$ from expert $\pi_0^h$ and train initial learner policy $\pi_0^r$.
2: **for** interactive deployment episode $i = 0 : M$ **do**
3:     Initialize $q_0^{lo}, q_0^{hi}$.
4:     **for** deployment timestep $t = 1 : H$ **do**
5:         Get predicted action label: $a_t^r \leftarrow \pi_i^r(x_t)$
6:         Construct calibrated uncertainty interval: $C_t(x_t) = [a_t^r - q_t^{lo}, a_t^r + q_t^{hi}]$.
7:         Compute robot query likelihood (e.g. $\mathbb{P}(\text{obs}_t^r \mid o_t; \tau) = 1$ if $u(x_t; \pi^r) > \tau$, else 0.)
8:         **if** robot queries (w.p. $\mathbb{P}(\text{obs}_t^r \mid o_t; \tau)$) or human intervenes (w.p. $\mathbb{P}(\text{obs}_t^h \mid o_t)$) **then**
9:             IQT update: $q_{t+1}^{lo,hi} \leftarrow q_t^{lo,hi} + \frac{\gamma_t}{p_t}(\text{err}_t - \alpha)\text{obs}_t$ w/ expert action $a_t^h \leftarrow \pi_i^h(x_t)$.
10:        **else**
11:            IQT update: $q_{t+1}^{lo,hi} \leftarrow q_t^{lo,hi}$
12:     Aggregated dataset with observed state and expert action pairs: $\mathcal{D}_{i+1} \leftarrow \mathcal{D}_i \cup \{(x, \pi_i^h(x))\}$
13:     Retrain learner: $\pi_{i+1}^r \leftarrow \arg\min_\pi \mathcal{L}(\mathcal{D}_{i+1})$

---

**Observation Model.** A key component of IQT is the observation likelihood model, $p_t = \mathbb{P}(\text{obs}_t = 1 \mid x_t)$. A nice byproduct of this model is that we can naturally derive a feedback model that is simultaneously *human-* and *robot-*gated by decomposing the observation model into the combination of both gating functions. Intuitively, the likelihood of observing the expert feedback at $t$ is given by the probability that the human chooses to give feedback *or* the robot asks the expert for feedback. Let $\text{obs}_t^h \in \{0, 1\}$ be a random variable representing observing a human-gated feedback (if $\text{obs}_t^h = 1$) where the expert initiates providing an action label $a_t^h = \pi^h(x_t)$ for $x_t$. Let $\text{obs}_t^r \in \{0, 1\}$ be a random variable representing robot-gated feedback, where the robot asks the expert for a label if $\text{obs}_t^r = 1$. We assume that if the robot elects to ask a question, the human will respond with probability 1. Our observation model takes the form: $p_t := \mathbb{P}(\text{obs}_t = 1 \mid x_t) = \mathbb{P}(\text{obs}_t^h = 1 \vee \text{obs}_t^r = 1 \mid x_t) = \mathbb{P}(\text{obs}_t^h = 1 \mid x_t) + \mathbb{P}(\text{obs}_t^r = 1 \mid x_t) - \mathbb{P}(\text{obs}_t^h = 1 \mid x_t)\mathbb{P}(\text{obs}_t^r = 1 \mid x_t)$. While our approach is not prescriptive about these models, in our experiments we model $\mathbb{P}(\text{obs}_t^h = 1 \mid x_t) = c$ as a small constant to represent infrequent human interventions. We describe the robot-gated likelihood model $\mathbb{P}(\text{obs}_t^r = 1 \mid x_t)$ below, informed by our uncertainty estimates.

**Quantifying Uncertainty: IQT.** In the interactive IL setting, IQT begins with initial upper and lower quantiles in the action (i.e. "label") space, $q_0^{lo}, q_0^{hi} \in \mathcal{A}$. The nonconformity score is a residual on the predicted ($a^r$) versus expert ($a^h$) action. Let $s_t^{lo}(a_t^r, a_t^h) = a_t^r - a_t^h$ be the lower residual (referred to as $s_t^{lo}$ for brevity), and $s_t^{hi}(a_t^r, a_t^h) = a_t^h - a_t^r$ be the upper residual (referred to as $s_t^{hi}$). If the expert action $a_t^h$ is observed, we compute the upper and lower miscoverage of $q_t^{lo}, q_t^{hi}$ as two indicator vectors: $\text{err}_t^{lo} = (s_t^{lo} < q_t^{lo})$ and $\text{err}_t^{hi} = (s_t^{hi} < q_t^{hi})$. IQT then updates the quantile estimates online to obtain $q_{t+1}^{lo}, q_{t+1}^{hi}$ via the update rule from equation [5](#). At the next timestep, the adjusted prediction interval is constructed with $C_{t+1}(x_{t+1}) = [a_{t+1}^r - q_{t+1}^{lo}, a_{t+1}^r + q_{t+1}^{hi}]$. If the expert action is not observed, $\text{obs}_t = 0$, then $q_{t+1}^{lo} = q_t^{lo}$ and $q_{t+1}^{hi} = q_t^{hi}$ and the prediction interval size remains the same.

Note that although $\text{err}_t$ is not known in the case where the expert does not provide an action label, IQT does not require it; IQT simply makes no change to the quantile estimate. In our simulated and hardware experiments, our action space, $q_0^{lo}$, and $q_0^{hi}$, are vector-valued. Our experiments instantiate IQT for continuous vectors, but the approach extends to discrete-valued action spaces.

**Leveraging Uncertainty: Asking for Help.** Finally, the calibrated intervals $C_{t+1}(x_{t+1})$ constructed by IQT enable us to design a new robot-gated feedback mechanism. Specifically, the model $\mathbb{P}(\text{obs}_t^r \mid x_t)$ is informed by the calibrated interval size, $u(x_{t+1}) := ||C_{t+1}(x_{t+1})||_2$. In simulation, we use $\mathbb{P}(\text{obs}_t^r \mid x_t) = \sigma(\beta[u(x_t; \pi^r) - \tau])$ where $\tau$ acts as an uncertainty threshold on $u(x_t; \pi^r)$, $\beta$ is a temperature hyperparameter, and $\sigma$ is the sigmoid function. In hardware experiments, we set $\tau$ as a hard threshold on on $u(x_t; \pi^r)$ above which $\mathbb{P}(\text{obs}_t^r \mid x_t) = 1$, and below which $\mathbb{P}(\text{obs}_t^r \mid x_t) = 0$.

**ConformalDAgger Algorithm.** We summarize ConformalDAgger in Algorithm 1, highlighting the difference from traditional DAgger (Ross et al., 2011). The learner interacts with the intermittent expert during $M$ interactive episodes. At the start of each, $q^{lo}, q^{hi}$ and are reset. Each episode lasts $H$ time steps and the intermittently-observed expert state-action pairs $(x_t, a_t^h)$ are aggregated with the prior training data in a fixed size training buffer to form the updated dataset, $\mathcal{D}_{i+1}$, which enables us to retrain the learner $\pi_{i+1}^r$ for the next deployment episode.

## 5 SIMULATED INTERATIVE IMITATION LEARNING EXPERIMENTS

To evaluate ConformalDAgger, we run a series of simulated experiments with access to an oracle expert. We ground our experiments in a simulated robot goal-reaching task (left, Figure 5) and study scenarios where the distribution shift occurs due to the expert's changing preference in goal location.

### 5.1 EXPERIMENTAL SETUP

**Task & Initial Learner Policy.** The robot learns a neural network policy $\pi^r : \mathcal{X} \to \mathcal{A}$ to move a cup from a start to the expert's desired goal location. We model input $x \in \mathcal{X} \subseteq \mathbb{R}^{12}$ as the xyz position of the robot as well as its binary gripper state (open or closed) across the previous 3 timesteps. The labels are actions, $a \in \mathcal{A} \subseteq \mathbb{R}^4$, represented as next xyz position and gripper state. The robot always starts at an initial $x$ with the gripper state closed (holding a cup) and must keep holding the cup while moving it to the unknown goal location that the expert prefers; let $\{g_0, g_1\}$ be two such goals. Before the first interactive deployment ($i = 0$), we simulate the expert as initially giving demonstrations placing the cup at $g_0$, where $\pi_{i=0}^h(x) = x + \omega \frac{g_0 - x}{\max_d(|g_0 - x|)}$, where $\max_d(\cdot)$ is a maximum over dimensions. The step size is regulated to be at most $\omega = 0.01$. The initial robot policy $\pi_{i=0}^r$ is trained on a dataset $\mathcal{D}_0$ of 10 expert trajectories with synthetically injected noise drawn from $\mathcal{N}(1, 0.5)$ for robustness as in (Laskey et al., 2017). See Appendix F for policy implementation details.

**Simulated Expert Policies.** To induce controlled distribution shift, we study three expert policies (left, column of Figure 5). (1) *Stationary*: the expert has a fixed goal, $g_0$, across all deployment episodes. (2) *Shift*: the expert goal shifts from $g_0$ to $g_1$ at deployment episode $i = 5$. For example, the expert may have decided that a different cup location is easier to reach. (3) *Drift*: expert's goal slowly drifts from $g_0$ to $g_1$ over the course of deployment episodes (in Figure 5 the drift from $g_0 \to g_{1a}$ occurs at episode $i = 5$, $g_{1a} \to g_{1b}$ occurs at episode $i = 8$, and $g_{1b} \to g_1$ occurs at episode $i = 11$). For example, the expert may start with a conservative goal location initially (e.g., a goal nearby) and incrementally move the cup closer and closer to their target goal that may be further out of reach. (4) *Environment Shift*: Under environment shift, the policy experiences covariate shift, where the deployment-time state distribution differs from the training distribution. Because the policy is trained on inputs which capture the robot's position, we instantiated environment shift by changing the starting position of the robot in the first episode, maintaining this new position for all latter episodes.

**Interactive Deployment Episodes & Learner Re-training.** We consider $M = 15$ deployment episodes before re-training the learner. Each deployment episode has two interactive task executions. The task ends when the cup has reached the correct goal position, $g^*$, or when the maximum timesteps (100) have been reached. The expert answers queries with optimal actions under their current policy $a^h = \pi_i^h(x)$. Following DAgger (Ross et al., 2011), after each deployment episode, the state-action pairs where the expert provided action labels are aggregated into the training dataset, forming aggregated buffer $\mathcal{D}_{i+1}$ for the next deployment episode $i + 1$. We constrain the size of the replay buffer to 300 datapoints, dropping the old experiences.

**Methods.** We compare **ConformalDAgger** to **EnsembleDAgger** (Menda et al., 2019), **LazyDAgger** (Hoque et al., 2021b), and **SafeDAgger** (Zhang & Cho, 2017). We performed a hyperparameter

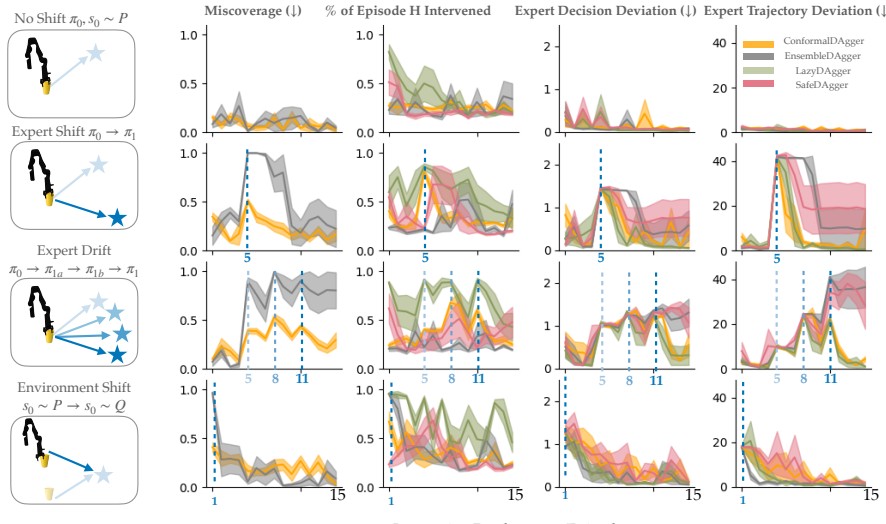

Figure 5: **Simulated Robot Results.** When the expert shifts or drifts (middle rows), **ConformalDAgger** increases the number of requests for expert feedback compared to **EnsembleDAgger** and **SafeDAgger**, but has less interventions than **LazyDAgger**. This decreases miscoverage and expert deviation after re-training. With a stationary expert (top) or environment shift (bottom), both **EnsembleDAgger** and **ConformalDAgger** perform similarly. Shading is std. error across 5 seeds.

sensitivity analysis for all methods (see B) and selected thresholds such that all approaches ask infrequent questions in the first few interactive deployment episodes, where no shift has occurred. **ConformalDAgger** uses an uncertainty threshold $\tau = 0.06$, temperature $\beta = 100$, lookback window $k = 100$, lr $= 0.6$, and initial $q_0^{lo,hi} = 0.01$. The uncertainty threshold $\tau$ is heuristically tuned to ask few, but infrequent questions in the first interactive deployment episode. **EnsembleDAgger** queries an expert online when there is high action prediction variance across an ensemble of learner policies, and when a safety classifier detects dissimilarity between expert and robot actions. We use 3 ensemble members and an uncertainty threshold of $\tau = 0.06$ for the ensemble disagreement and a safety classifier threshold of $s = 0.03$. **LazyDAgger** uses $s = 0.03$ to begin human intervention, and only switches back to autonomous mode when the deviation between the learner's prediction and expert's action are below a context-switching threshold, $0.1 * s$. To make **SafeDAgger**'s number of initial interventions comparable, we decrease the safety classifier threshold to $s = 0.01$.

**Metrics.** We measure the quality of our uncertainty quantification via the **miscoverage rate** and human effort via the **intervention percentage** of the deployment trajectory. We compute the miscoverage for EnsembleDAgger using three times the standard deviation as the prediction interval. We measure the quality of the learned policy via two metrics. **Decision deviation** simulates the learner's behavior under its current policy and queries the expert at each learner state to obtain an expert action label. We measure the average L2 distance between the predicted and expert action. **Trajectory deviation** forward simulates both the expert and the learner acting independently under their policies, starting from the same initial $x$. We compare the L2 distance between the trajectories.

## 5.2 EXPERIMENTAL RESULTS

We focus results on infrequent human-gated feedback where $\mathbb{P}(\text{obs}_t^h = 1 \mid x_t) = 0.2, \forall t$. In Appendix F, we present experiments with frequent (=0.9) and partial (=0.5) feedback. Irrespective of the human-gated likelihood, the learner can always *actively* ask for feedback based on $\mathbb{P}(\text{obs}_t^r \mid x_t)$.

**Takeaway 1: When the expert *shifts*, ConformalDAgger asks for more help immediately compared to EnsembleDAggerand SafeDAgger, enabling the algorithm to more quickly align to the expert with fewer human interventions than LazyDAgger.** Consider deployment episode $i = 5$ where the expert shifts from goal 0 to goal 1 (center row, Figure 5). ConformalDAgger immediately increases the number of expert feedback requests: before shift, the expert intervened $\sim$20% of the time but in the 5th episode they intervene $\sim$60% of the time. In contrast, EnsembleDAgger remains close to $\sim$30%. Relatedly, ConformalDAgger has consistently lower miscoverage rate (max = 0.4 at shift; converges to 0.2) compared to the baseline (max = 1.0 at shift, converges to 0.4). Due to

the extra solicited feedback, ConformalDAgger's ultimate decision and trajectory deviations are minimized in the subsequent retrained policies.

**Takeaway 2: ConformalDAgger automatically asks for more help each time the expert *drifts*.** In the third row of Figure 5, we see ConformalDAgger maintain a similar level of queries as EnsembleDAgger during the first shift (at episode 5) and last shift (episode 11) but increases feedback during the intermediate shift (at episode 8). Despite these similarities, EnsembleDAgger's miscoverage rate is consistently higher than ConformalDAgger's and EnsembleDAgger's re-trained policy on average does not adapt as quickly (with higher expert decision and trajectory deviation). We hypothesize that this is because ConformalDAgger asks frequent questions consistently across all seeds compared to EnsembleDAgger (i.e. lower variance in Fig 5).

**Takeaway 3: With a *stationary* expert, ConformalDAgger and baselines are similar.** We find that without distribution shift (top row, Figure 5), all methods ask for minimal help (staying near the 20% human-gated probability), with **LazyDAgger** asking slightly more questions, and achieve similar performance in alignment to the expert.

**Takeaway 4: Under *environment shift*, ConformalDAgger is comparable to baselines.** In the fourth row of Figure 5, we see that **EnsembleDAgger** and **ConformalDAgger** balance intervention frequency with policy learning quality marginally better than **LazyDAgger** or **SafeDAgger**.

## 6 HARDWARE EXPERIMENTS

Finally, we deployed **ConformalDAgger** in hardware on a 7 degree-of-freedom robotic manipulator that uses a state-of-the-art Diffusion Policy (Chi et al., 2023) trained via IL to perform a sponging task (Figure 1). The goal of our hardware experiments is to demonstrate how our approach can scale to a high-dimensional, real-world policy and understand how ConformalDAgger enables the robot learner to query a real human teleoperator. Our goal is to train the robot to perform a real-world cleaning task where it wipes up a line drawn by an Expo marker on a whiteboard with sponge.

**Human Expert.** The human expert teleoperates the robot via a Meta Quest 3 remote controller. They initially provide 50 demonstrated trajectories moving in a straight-line path along the marker line to construct the initial training dataset, $\mathcal{D}_0$. We model human-gated interventions with probability $\mathbb{P}(\text{obs}_t^h = 1 \mid x_t) = 0.2$, $\forall t$. To control this rate, our interface queries the user with 20% probability at the any timepoint along deployment to simulate that consistent independent intervention rate. We induce a potential distribution shift during interactive deployment episodes by changing the expert's wiping strategy from a straight line to a zig-zag pattern.

**Learner Policy.** We represent the robot's policy as a CNN-based diffusion policy (Chi et al., 2023). The policy predicts $a^r \in \mathcal{A} \equiv \mathcal{Y}$ as 16 future actions, where each action is an end-effector position and quaternion orientation. The inputs $x \in \mathcal{X}$ are the current and previous image observations from the wrist and third-person camera. Image observations are encoded using a ResNet-18 visual encoder (trained end-to-end with the diffusion policy) and the action-generating process is conditioned on encoded observation features with FiLM (Perez et al., 2018). The initial policy $\pi_0^r$ is trained for 50K iterations (training parameters in Appendix Sec H.1). The robot is deployed to interactively execute the task 25 times after which its policy is retrained. This is an interactive deployment where the robot queries the user via our ConformalDAgger algorithm, or the user can independently intervene intermittently. During expert feedback, the human teleoperates the robot for a sequence of 16 timesteps, after which, if the robot doesn't ask for feedback, control is handed back over to the robot. After the interactive deployment, the initial learner policy is fine-tuned for an additional 35K iterations. We reset the learning-rate schedule, aggregate datapoints at which the expert gave feedback, and cap the training buffer size at 6k most recent input-action pairs.

**ConformalDAgger Hyperparameters.** We apply IQT on only the predicted end-effector position, because we reason about nonconformity via the signed residual in Euclidean space. For quaternion rotation, we calibrate only the positional output to avoid the complexity of multiple IQT processes for positional and rotational nonconformities.

We initialize $q_0^{lo}, q_0^{hi} = 0.01$, our desired coverage level is $\alpha = 0.1$, and $\gamma_t = 0.15 \hat{B}_t$, where the lookback window for $\hat{B}_t$ is 20. We use a simple model for $\mathbb{P}(\text{obs}^r \mid x_t) = 1$ that asks for help if the L2 norm of the uncertainty interval $C_t(x_t)$ exceeds threshold $\tau = 0.07$.

**Baseline.** Because the diffusion policy implicitly represents the action distribution present in the training data, we sample from the policy N=3 times and evaluate variance over the predicted action in

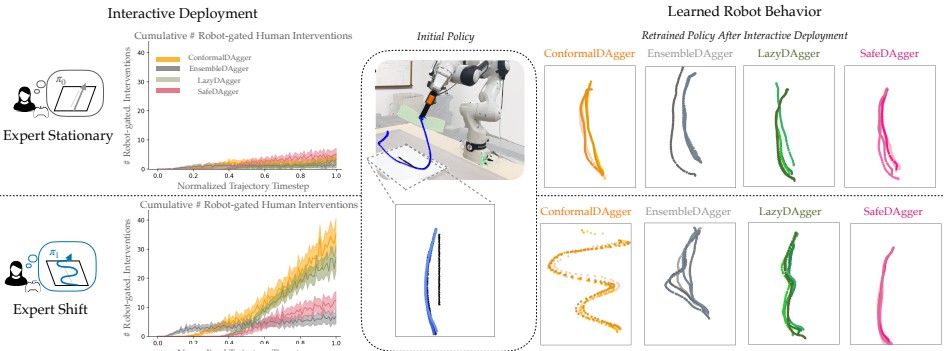

Figure 6: **Hardware Results.** (bottom, left) Conformal enables the robot to significantly increase the number of questions when the expert shifts their strategy while Safe and Ensemble only increase slightly. (bottom, right) Qualitatively, Conformal learns a higher quality zigzag strategy. Lazy exhibits a similar an intervention profile to Conformal, but because the algorithm doesn't ask for help consistently in each episode, it does not receive enough additional data to learn the shifted policy.

order to capture the notion of epistemic uncertainty. We consider this an *implicit* EnsembleDAgger: we use the implicit ensemble approach from (Wolleb et al., 2022) by sampling multiple times from the same diffusion model. The safety classifier is a fully connected network that takes as input the ResNet-18 embeddings of the camera images concatenated with robot proprioceptive state. We set the classifier threshold $s$ to be the mean of the distribution of expert-learner divergence over the training datapoints. EnsembleDAgger's uncertainty threshold is 0.005, at approximately the mean of the distribution of ensembles variances (over position and orientation predicted by the policy) over the training data, tuned such that the learner asks occasional but not excessive questions. We also compared with **LazyDAgger** (threshold of $0.25 * s$, informed by Hoque et al. (2021a)), and **SafeDAgger**. All models use the same initial learner policy.

**Results.** The left of Figure 6 shows quantitative results for both methods. **ConformalDAgger** increases the cumulative number of robot-requested interventions up to about 35 timesteps of human feedback when interacting with the *shifted* expert (who switches to a zig-zag pattern) compared to the *stationary* expert (who always wipes in a straight line) *which does not induce as frequent of questions*. To better understand this, we analyzed Figure 1, which shows **ConformalDAgger** engaging with each type of expert in one interactive deployment episode. Here, we see how ConformalDAgger has uncertainty when interacting with the stationary expert (due to inherent noise in the VR-teleoperated intermittent human interventions) but the robot's uncertainty never rises above the threshold the robot *initiates* expert feedback. This is in stark contrast to the expert shift scenario (right of Figure 1) where half-way through the deployment episode the uncertainty intervals become large enough to trigger robot asking for help. On the other hand, with the shifted expert, **EnsembleDAgger** and **SafeDAgger** do not request as many interventions as **LazyDAgger** and **ConformalDAgger**, (Figure 6). When **LazyDAgger** asks for help the queries follow a similar profile as **ConformalDAgger**, but it does not reliably ask every episode, leading to less data for learning the shifted strategy. We hypothesize that this may be because the training demonstrations have low variance while the sponge is in contact with the whiteboard, but higher variance as the robot approaches the table. Because **EnsembleDAgger**'s and **SafeDAgger**'s uncertainty is uncalibrated to the deployment-time expert's data, it is unable to identify the need for additional feedback when it reaches the whiteboard. Qualitative results are shown on the right of Figure 6. As expected, all approaches remain aligned with the *stationary* expert when retrained. However, under the *shifted* expert, rollouts from the retrained **ConformalDAgger** policy exhibit a more distinct zig-zag pattern, compared to **EnsembleDAgger** and **SafeDAgger**, likely because the approaches did not ask for as much help.

## 7 CONCLUSION

We first extend uncertainty quantification via online conformal prediction to handle intermittent labels, such as those observed in interactive imitation learning. We then propose ConformalDAgger, a unification of our online conformal prediction algorithm with interactive imitation learning. Our approach provides asymptotic coverage guarantees for deployed end-to-end policies, uses the calibrated uncertainty measure to detect expert distribution shifts and actively query for more feedback, and empirically enables the robot learner update its policy to better align with the expert.

ACKNOWLEDGMENTS

The authors would like to thank Gokul Swamy for insightful conversations and the detailed review, Yilin Wu for help with diffusion policy and robot hardware setup. MZ is supported by an NDSEG fellowship.

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
