APPENDIX

## A    ADDITIONAL BACKGROUND ON CONFORMAL PREDICTION FOR ROBOTICS

**Conformal Prediction for Robotics.** Recently, conformal prediction has become popular in the robotics domain in part due to the distribution-free guarantees it provides for arbitrarily complex learned models present within modern robotics pipelines. Specifically, conformal prediction has been used to provide collision-avoidance assurances (Chen et al., 2021; Lindemann et al., 2023; Dixit et al., 2023; Muthali et al., 2023; Taufiq et al., 2022; Dietterich & Hostetler, 2022; Lin & Bansal, 2024), calibrate early warning systems (Luo et al., 2022), and quantify uncertainty in large language model based planners (Ren et al., 2023; Lidard et al., 2024). There are several core challenges with the input-and-label data encountered in robotics: data is non-i.i.d. (e.g., sequential decision-making), data distributions are non-stationary (e.g. changing environment conditions), and labels are intermittently observed (e.g., limited expert feedback in the IL domain). By extending online conformal prediction to the intermittent label setting, we take a step towards addressing these challenges.

## B    HYPERPARAMETER SENSITIVITY ANALYSIS

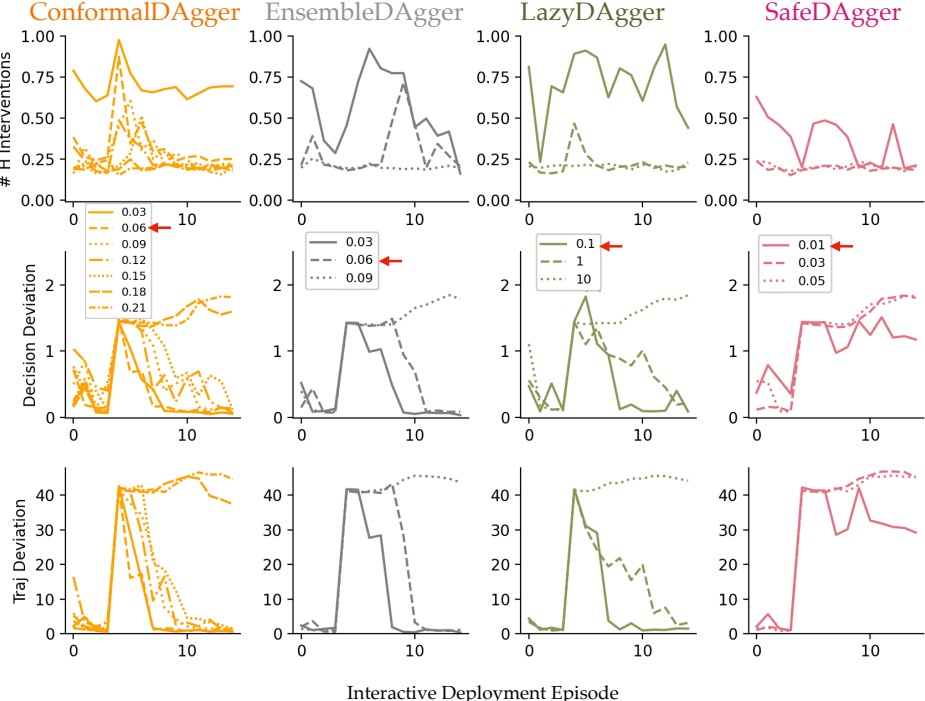

Figure 7: Hyperparameter sensitivity analysis under Expert Shift (Human-gated probability $P(obs_t^h|o_t) = 0.2$).

We show below a sensitivity analysis in the Expert Shift (Human-gated probability $P(obs_t^h|o_t) = 0.2$) context of the uncertainty threshold for **ConformalDAgger** and **EnsembleDAgger**, where we vary the uncertainty threshold between 0.03 and 0.09 (Figure 7). To demonstrate the results over the deployment episodes, we hold the **EnsembleDAgger** safety threshold constant at 0.03. The results are averaged over 3 seeds.

When the threshold is low (0.03), the approaches all identify the shift and ask for help accordingly; however, this comes at the cost of a great deal of human feedback requested throughout every deployment episode. Even in the first episode in which there has been no expert shift, a low-threshold at 0.03 requires conformal and ensemble to ask questions 70-80% of the time. When the threshold is set too high, at 0.09, **EnsembleDAgger** is unable to identify the shift and continuously mispredict.

We included in our sensitivity analysis **LazyDAgger** 's context-switching threshold, where we tested values of 0.1, 1, and 10, which are thresholds presented in the **LazyDAgger** sensitivity analysis in Hoque et al. (2021b). Similarly, we hold the **LazyDAgger** safety threshold constant at 0.03. For **SafeDAgger**, we examined safety thresholds of 0.01, 0.03, and 0.05.

Across all of the algorithms, ConformalDAgger's expert deviation is the least prone to threshold selection, where all models are able to effectively adapt to the expert's shifted policy. Empirically, we find that if $\tau$ is set too high (at 0.18, 0.21), ConformalDAgger may not flag high uncertainty, relying instead on human-gated interventions to update IQT. When we ablate the values of tau between the values [0.03, 0.06, 0.09, 0.12, 0.15], we found ConformalDAgger to be fairly robust in terms of miscoverage (see updated Figure 7 in Appendix B, $\tau$=0.18, 0.21 included as well).

Theoretically, if $\tau$ is set too high, failing to properly reflects the distribution of encountered misprediction residuals, ConformalDAgger will not flag high uncertainty, relying on human-gated interventions only ($P(obs_t^h|o_t)$) to update IQT and provide retraining data, which may result in poor coverage. However, importantly, one benefit of IQT is that its theoretical guarantees afford us an understanding of what the miscoverage gap will be. Recall that Proposition 1 gives us a bound on the miscoverage gap between the achieved miscoverage rate and desired miscoverage rate $\alpha$. This gap depends on 1) the time horizon $T$ that we are running our algorithm, and 2) the maximum value of $\frac{\gamma_t}{p_t}$. In cases where $\tau$ is set too high such that ($P(obs_t^r|o_t)$ is close to 0), meaning the robot almost never queries for additional help, the probability of getting feedback $p_t$ is approximately the $P(obs_t^h|o_t)$, relying only on the likelihood of human feedback. For short sequences ($T$ small), and infrequent human-gated feedback ($p_t$ small), Proposition 1 informs us that this miscoverage gap might be large. Proposition 1 shows the theoretical basis for that when human-gated feedback is infrequent, and highlights that a proper and effective selection of $\tau$ to raise $P(obs_t^r|o_t)$ and subsequently $p_t$, helps to reduce the miscoverage gap.

The red arrows indicate the threshold values we ended up using for our 4D reaching experiments. We used the number of human interventions requested as a proxy metric for helping us choose the thresholds. Although the **EnsembleDAgger** threshold of 0.03 more quickly learns the expert policy than 0.06, however, the 0.03 threshold causes the robot to asks many more questions at the initial episode (even without the existence of shift). We chose the threshold of 0.06 for **EnsembleDAgger** and **ConformalDAgger** because in the first 4 episodes before the shift occurs, the algorithm asks some, but not excessive, questions. **LazyDAgger** uses the safety classifier prediction, $s = 0.03$ to begin human intervention, and only switches back to autonomous mode when the learner's prediction and human ground truth action received online are below a context-switching threshold, $0.1s$. Because **SafeDAgger** alone under $s = 0.03$ does not ask as many questions as **EnsembleDAgger** and **LazyDAgger** which both involve additional mechanisms for querying alongside the safety classifier, we choose $s = 0.01$ for **SafeDAgger**.

## C  PROOF OF PROPOSITION 1

We will start by proving the following lemma. Recall that $B$ is the upper bound on $q_1 \in [0, B]$, and $s_t \in [0, B]$ by definition.

**Lemma 1.** *For all $t$, we have $-\alpha N_{t-1} \leq q_t \leq B + (1 - \alpha)N_{t-1}$, where $N_t = \max_{1 \leq r \leq t} \frac{\gamma_r}{p_r}$. $B$ is the upper bound on $q_1 \in [0, B]$ and $s_t \in [0, B]$.*

*Proof.* $q_1 \in [0, B]$ by assumption, so the lemma is satisfied at $t = 1$. Assume $q_t \in [-\alpha N_{t-1}, B + (1 - \alpha)N_{t-1}]$. Now, for $q_{t+1}$, we consider Case 1, where $obs_t = 0$, which means that $q_{t+1} = q_t$, so $q_{t+1}$ lies within the range $[-\alpha N_{t-1}, B + (1 - \alpha)N_{t-1}]$. Since $N_t \geq N_{t-1}$, $q_{t+1}$ also lies within the larger range $[-\alpha N_t, B + (1 - \alpha)N_t]$, as desired. In Case 2 where $obs_t = 1$, $q_{t+1} = q_t + \frac{\gamma_t}{p_t}(\text{err}_t - \alpha)$. If we represent $\eta_t := \frac{\gamma_t}{p_t}$, we obtain the constant-feedback quantile tracking update: $q_{t+1} = q_t + \eta_t(\text{err}_t - \alpha)$, with variable $\eta_t$ instead of $\gamma_t$. Then, Lemma 1 in Angelopoulos et al. (2024), with $N_t = \max_{1 \leq r \leq t} \eta_t$ bounds $q_{t+1}$ within the range $[-\alpha N_t, B + (1 - \alpha)N_t]$. □

Next, we proceed to show how we can leverage the existing quantile tracking results to derive our results with intermittent feedback.

*Proof.* First, we take expectations only with respect to $\mathrm{obs}_t$ conditional on $x_t$, $\mathrm{err}_t$, and all other randomness, noting in particular that $\mathrm{obs}_t$ is independent of everything else given $x_t$ and has a Bernoulli distribution with mean $p_t$.

We will abbreviate the left-hand side by $\mathbb{E}[q_{T+1}|D_{T+1}]$, where $D_{T+1} := \{\mathrm{err}_t, x_t\}_{t \leq T+1}$, and use $\mathbb{E}_{\mathrm{obs}_{1:t}} = \mathbb{E}_{\mathrm{obs}_1 \sim p_1, \ldots, \mathrm{obs}_t \sim p_t}$ to denote this conditional expectation for brevity below:

$$\mathbb{E}_{\mathrm{obs}_{1:t}}[q_{T+1} - q_r | D_{T+1}] = \mathbb{E}_{\mathrm{obs}_{1:t}}[q_{T+1}|D_{T+1}] - \mathbb{E}_{\mathrm{obs}_{1:r}}[q_r|D_r] \tag{7}$$

$$= \mathbb{E}_{\mathrm{obs}_{1:t}}[q_1 + \sum_{t=1}^{T} \frac{\gamma_t}{p_t}(\mathrm{err}_t - \alpha)\mathrm{obs}_t | D_{T+1}] - \mathbb{E}_{\mathrm{obs}_{1:r}}[q_1 + \sum_{t=1}^{r} \frac{\gamma_t}{p_t}(\mathrm{err}_t - \alpha)\mathrm{obs}_t | D_r] \tag{8}$$

$$= q_1 + \sum_{t=1}^{T} \frac{\gamma_t}{p_t}(\mathrm{err}_t - \alpha)p_t - \left(q_1 + \sum_{t=1}^{r} \frac{\gamma_t}{p_t}(\mathrm{err}_t - \alpha)p_t\right) \tag{9}$$

$$= \sum_{t=r}^{T} \gamma_t(\mathrm{err}_t - \alpha). \tag{10}$$

Given the definition of $\Delta$, we have that $\gamma_t^{-1} = \sum_{r=1}^{t} \Delta_r$ for all $t \geq 1$. So

$$\left|\frac{1}{T}\sum_{t=1}^{T}(\mathrm{err}_t - \alpha)\right| = |\frac{1}{T}\sum_{t=1}^{T}(\sum_{r=1}^{t}\Delta_r)\gamma_t(\mathrm{err}_t - \alpha)| \tag{11}$$

$$= \left|\frac{1}{T}\sum_{r=1}^{T}\Delta_r(\sum_{t=r}^{T}\gamma_t(\mathrm{err}_t - \alpha))\right| \tag{12}$$

$$= \left|\frac{1}{T}\sum_{r=1}^{T}\Delta_r\left(\mathbb{E}_{\mathrm{obs}_{1:t}}[q_{T+1} - q_r|D_{T+1}]\right)\right|. \tag{13}$$

By Lemma 1, this expected difference is bounded by $B + (1-\alpha)N_T - (-\alpha N_T)$:

$$\mathbb{E}_{\mathrm{obs}_{1:T+1}}[q_{T+1} - q_r | D_{T+1}] \leq B + \max_{1 \leq t \leq T} \frac{\gamma_t}{p_t}. \tag{14}$$

In other words, we can drop the expectation via Lemma 1 and consider the worst case bound on $q_{T+1} - q_r$. Thus,

$$\left|\frac{1}{T}\sum_{t=1}^{T}\mathrm{err}_t - \alpha\right| \leq \frac{1}{T}\sum_{r=1}^{T}|\Delta_r|\left(B + \max_{1 \leq t \leq T}\frac{\gamma_t}{p_t}\right) \tag{15}$$

$$= \frac{1}{T}||\Delta_{1:T}||_1\left(B + \max_{1 \leq t \leq T}\frac{\gamma_t}{p_t}\right). \tag{16}$$

This completes the proof. $\square$

### C.1 SPECIAL CASE: WHEN $\gamma_t = p_t$

Next, given the quantile tracking update with an intermittent observation model, we consider what would happen if $\gamma_t$ was set as $p_t$. When $\gamma_t = p_t$, then quantile tracking update becomes $q_{t+1} = q_t + (\mathrm{err}_t - \alpha)\mathrm{obs}_t$ and $\max_{1 \leq t \leq T} \frac{\gamma_t}{p_t} = 1$.

Under Proposition 1, IQT with $\gamma_t = p_t$ gives the following finite time coverage bound:

$$\left|\frac{1}{T}\sum_{t=1}^{T}\mathrm{err}_t - \alpha\right| \leq \frac{B+1}{T}||\Delta_{1:T}||_1 \tag{17}$$

where the sequence $\Delta$ is defined with values $\Delta_1 = \gamma_1^{-1}$ and $\Delta_t = \gamma_t^{-1} - \gamma_{t-1}^{-1}$ for all $t \geq 2$.

## D INTERMMMITTENT ADAPTIVE CONFORMAL INFERENCE

We show in this section that intermittent observation of ground truth labels can be extended to Adaptive Conformal Inference (ACI) (Gibbs & Candes, 2021). To facilitate understanding, we briefly summarize ACI and discuss our extension Intermittent Adaptive Conformal Inference (IACI).

**Setup: Quantile Regression (QR).** Similar to IQT, consider an *arbitrary* sequence of data points $(x_t, y_t) \in \mathcal{X} \times \mathcal{Y}$, for $t = 1, 2, ...$, that are not necessarily I.I.D. Our goal in ACI is to also produce prediction sets on the output of any base prediction model such that the sets contain the true label with a specified miscoverage rate $\alpha$. Mathematically, at each time $t$, we observe $x_t$ and seek to cover the true label $y_t$ with a set $C_t(x_t)$, which depends on a base prediction model, $\hat{f} : \mathcal{X} \rightarrow \mathcal{Y}$. We will discuss ACI with a conformal quantile regression Romano et al. (2019) backbone. The base model takes as input the current $x_t$ and outputs prediction $\hat{y}_t$ as well the *estimated upper and lower conditional quantiles*:

$$\{\hat{q}_{\alpha_{lo}}(x_t), \hat{y}_t, \hat{q}_{\alpha_{hi}}(x_t)\} \leftarrow \hat{f}(x_t), \quad \forall (x_t, y_t), \tag{18}$$

where $\hat{q}_{\alpha_{lo}}(x_t)$ is an estimate of the $\alpha_{lo}$-th conditional quantile and $\hat{q}_{\alpha_{hi}}(x_t)$ is the $\alpha_{hi}$-th quantile estimate. During training, $\hat{q}_{\alpha_{lo}}(x_t)$ is learned with an additional Pinball loss Koenker & Bassett Jr (1978); Romano et al. (2019).

**Adaptive Conformal Inference (ACI).** At each time $t$, we compute the nonconformity score $s_t$ as:

$$s(x_t, y_t; f) = \max\{\hat{q}_{\alpha_{lo}}(x_t) - y_t, y_t - \hat{q}_{\alpha_{hi}}(x_t)\}, \tag{19}$$

which is the coverage error induced by the regressor's quantile estimates.

Let $S_t$ be the set of conformity scores for all data points through time $t$ in $\mathcal{D}_{\text{calib}}$. In general, the magnitude of $s(x_t, y_t; \hat{f})$ is determined by the miscoverage error and its sign is determined by if the true value of $y_t$ lies outside or inside the estimated interval.

Mathematically, the calibrated prediction interval for $Y_t$ is:

$$C_t(x_t) = \big[\hat{q}_{\alpha_{lo}}(x_t) - \hat{Q}_{S_t}(1 - \alpha_t), \tag{20}$$

$$\hat{q}_{\alpha_{hi}}(x_t) + \hat{Q}_{S_t}(1 - \alpha_t)\big], \tag{21}$$

where the $\hat{Q}_{S_t}(1 - \alpha_t) := (1 - \alpha_t)(1 + \frac{1}{|\mathcal{D}_{\text{calib}}|})$-th adaptive empirical quantile of $S_t = S_{t-1} \cup s(x_t, y_t; \hat{f})$. The empirical quantile is defined as the following (where $k$ is the lookback window):

$$\hat{Q}_{S_t}(c) := \inf\left\{m : \left(\frac{1}{|D_{t-k:t}|} \sum_{(x_i, y_i) \in D_{t-k:t}} \mathbb{1}_{\{s(X_i, Y_i) \leq m\}}\right) \geq c\right\} \tag{22}$$

Given the non-stationarity of the data distribution, ACI examines the empirical miscoverage frequency of the previous interval, and then decreases or increases a time-dependent $\alpha_t$. Fixing step size parameter $\gamma > 0$, ACI updates

$$\alpha_{t+1} := \alpha_t + \gamma(\alpha - \text{err}_t) \tag{23}$$

**ACI Coverage Guarantee.** The adaptive quantile adjustments made in ACI provide the following coverage guarantee:

$$\left|\frac{1}{T} \sum_{t=1}^{T} \text{err}_t - \alpha\right| \leq \frac{\max\{\alpha_1, 1 - \alpha_1\} + \gamma}{T\gamma} \tag{24}$$

obtaining the desired $\alpha$ coverage frequency without making an assumptions on the data-generating distribution (Proposition 4.1 in Gibbs & Candes (2021)). As $T$ approaches $\infty$, $\lim_{T \to \infty} \frac{1}{T} \sum_{t=1}^{T} \text{err}_t$ approaches $\alpha$. This guarantees ACI gives the $1 - \alpha$ long-term empirical coverage frequency regardless of the underlying data generation process.

## D.1 INTERMITTENT ADAPTIVE CONFORMAL PREDICTION

To achieve Intermittent Adaptive Conformal Inference (IACI), we update $\alpha_t$ at each timestep $t$ with Equation 25:

$$\alpha_{t+1} = \alpha_t + \frac{\gamma}{p_t}(\alpha - \text{err}_t)\text{obs}_t \tag{25}$$

Recall $\text{err}_t = \mathbb{1}_{y_t \notin C_t(x_t)}$ and $\text{obs}_t$ represents whether $y_t$ was observed at timestep $t$, and $p_t = \mathbb{P}(\text{obs}_t = 1 | x_t) \in (0, 1]$. The calibrated prediction interval for $y_t$ becomes:

$$C_t(x_t) = [\hat{q}_{\alpha_{lo}}(x_t) - \hat{Q}_{S_t^{obs}}(1 - \alpha_t), \hat{q}_{\alpha_{hi}}(X_t) + \hat{Q}_{S_t^{obs}}(1 - \alpha_t)], \tag{26}$$

where $\hat{Q}_{S_t^{obs}}(1 - \alpha_t) := (1 - \alpha_t)(1 + \frac{1}{|\mathcal{D}_{\text{calib}}|})$-th adaptive empirical quantile of $S_t^{obs}$, the set of nonconformity scores that have been observed, weighted by their probability of observation. Since we will only observe feedback with probability $p_t$ at each timestep, our set of nonconformity values will not be the full set of scores at every timestep. Instead, we have access to some subset of nonconformity scores $S_t^{obs}$, where $p_i$ is the probability of element $s(x_i, y_i; f)$ being in the subset $S_t^{obs}$ for $1 \leq i \leq t$. We define $\hat{Q}_{S_t^{obs}}(c)$:

$$\hat{Q}_{S_t^{obs}}(c) := \inf \left\{ m : \left( \frac{1}{|D_{t-k:t}|} \sum_{(x_i, y_i) \in D_{t-k:t}} \frac{1}{p_i} \cdot \text{obs}_i \cdot \mathbb{1}_{\{s(x_i, y_i) \leq m\}} \right) \geq c \right\} \tag{27}$$

At best, we can say that in expectation, the inclusion criteria for any element in the summation term is equivalent for the summation in $\hat{Q}_t^{obs}(x)$ and the summation in $\hat{Q}_t(x)$: $\mathbb{E}_{\text{obs}_i \sim p_i}[\frac{1}{p_i} \cdot \text{obs}_i \cdot \mathbb{1}_{\{s(X_i, Y_i) \leq m\}}] = \mathbb{1}_{\{s(X_i, Y_i) \leq m\}}$.

**Proposition 2.** *IACI Coverage Guarantee. With this intermittent feedback update, with probability one, where $M_t = \min \{p_1, ..., p_t\} \in (0, 1]$, we have that for all $T \in \mathbb{N}$:*

$$\left| \frac{1}{T} \sum_{t=1}^{T} \text{err}_t - \alpha \right| \leq \frac{\max\{\alpha_1, 1 - \alpha_1\} + \frac{\gamma}{M_{T+1}}}{T\gamma} \tag{28}$$

At $\lim_{T \to \infty} \frac{1}{T} \sum_{t=1}^{T} \text{err}_t$ approaches $\alpha$, given $M_\infty = \min \{p_1, ..., p_\infty\} \in (0, 1]$. This guarantees IACI gives the $1 - \alpha$ long-term empirical coverage frequency regardless of the underlying data generation process. The proof follows from Gibbs & Candes (2021) as is described in the next subsection (Section D.2).

## D.2 Proof for Intermittent Adaptive Conformal Inference

**Assumptions.** We will assume throughout that with probability one, $\alpha_1 \in [0, 1]$, $\alpha \in (0, 1)$, $p_t \in (0, 1] \; \forall 1 \leq t \leq \infty$, and $\hat{Q}_t^{obs}(x)$ is non-decreasing with $\hat{Q}_{S_t^{obs}}(c) = -\infty$ for all $c < 0$, and $\hat{Q}_{S_t^{obs}}(c) = \infty$ for all $c > 1$.

**Lemma 2.** *With probability one, we have that $\forall t \in [1, T]$,*

$$\alpha_t \in [-\frac{\gamma}{M_T}, 1 + \frac{\gamma}{M_T}] \tag{29}$$

*where $M_T = \min_{1 \leq r \leq T} p_r$.*

*Proof.* Observe that

$$\sup_{1 \leq t \leq T} |\alpha_{t+1} - \alpha_t| = \sup_{1 \leq t \leq T} |\frac{\gamma}{p_t}(\alpha - \text{err}_t)| \tag{30}$$

$$\leq \sup_{1 \leq t \leq T} \frac{\gamma}{M_T} |(\alpha - \text{err}_t)| = \frac{\gamma}{M_T} \sup_{1 \leq t \leq T} |(\alpha - \text{err}_t)| \tag{31}$$

$$< \frac{\gamma}{M_T} \tag{32}$$

The rest of the proof follows from Lemma 4.1 in Gibbs & Candes (2021). We will write it out explicitly here.

**Lower Bound.** Assume towards contradiction that with positive probability the set $\{\alpha_t\}_{t \in [1,T]}$ is such that $\inf_{t \in [1,T]} \{\alpha_t\} < -\frac{\gamma}{M_T}$, which means there exists some element in the set $\{\alpha_t\}_{t \in [1,T]}$

such that the element is less than $-\frac{\gamma}{M_T}$. Let $\alpha_{t+1}$ be the first element in set $\{\alpha_t\}_{t\in[1,T]}$ such that $\alpha_{t+1} < -\frac{\gamma}{M_T}$.

We know by assumption that $\alpha_1 \in [0,1]$, so $\alpha_{t+1}$ cannot be the first value. So $\alpha_t > \alpha_{t+1}$, else the latter would not be the first element such that $\alpha_{t+1} < -\frac{\gamma}{M_T}$. Since we know that $\alpha_t - \alpha_{t+1} < \frac{\gamma}{M_T}$ (from the observation above), then

$$\alpha_t - \alpha_{t+1} < \frac{\gamma}{M_T} \tag{33}$$

$$\Leftrightarrow \quad \alpha_t < \frac{\gamma}{M_T} + \alpha_{t+1} \tag{34}$$

$$\Leftrightarrow \quad \alpha_t < \frac{\gamma}{M_T} + \alpha_{t+1} < \frac{\gamma}{M_T} + (-\frac{\gamma}{M_T}) \tag{35}$$

$$\Leftrightarrow \quad \alpha_t < 0. \tag{36}$$

However, if $\alpha_t < 0$, then $\hat{Q}_{S_t^{obs}}(1-\alpha_t) = \infty \Rightarrow \mathrm{err}_t = 0$. This is because the quantile will be the trivially large infinite quantile, meaning there will definitely be no undercoverage at $t$. Since $\frac{\gamma}{p_t}\alpha$ is positive by definition,

$$\Rightarrow \alpha_{t+1} = \alpha_t + \frac{\gamma}{p_t}(\alpha - \mathrm{err}_t) = \alpha_t + \frac{\gamma}{p_t}(\alpha - 0) \geq \alpha_t \tag{37}$$

We have reached a contradiction with $\alpha_{t+1}$ being the first infimum value reached.

**Upper Bound.** The upper bound argument is symmetric but we will write it out explicitly. Assume towards contradiction that with positive probability the set $\{\alpha_t\}_{t\in[1,T]}$ is such that $\sup_{t\in[1,T]}\{\alpha_t\} > 1 + \frac{\gamma}{M_T}$, which means there exists some element in the set $\{\alpha_t\}_{t\in[1,T]}$ such that the element is greater than $1 + \frac{\gamma}{M_T}$. Let $\alpha_{t+1}$ be the first element in set $\{\alpha_t\}_{t\in[1,T]}$ such that $\alpha_{t+1} > 1 + \frac{\gamma}{M_T}$.

We know by assumption that $\alpha_1 \in [0,1]$, so $\alpha_{t+1}$ cannot be the first value. So there exists some $\alpha_t < \alpha_{t+1}$, where $\alpha_{t+1} > 1 + \frac{\gamma}{M_T}$. Since we know that $\alpha_{t+1} - \alpha_t < \frac{\gamma}{M_T}$ (from the observation above), then

$$\alpha_{t+1} - \alpha_t < \frac{\gamma}{M_T} \tag{38}$$

$$\Leftrightarrow \quad -\alpha_t < \frac{\gamma}{M_T} - \alpha_{t+1} \tag{39}$$

$$\Leftrightarrow \quad \alpha_t > \alpha_{t+1} - \frac{\gamma}{M_T} > (1 + \frac{\gamma}{M_T}) - \frac{\gamma}{M_T} \tag{40}$$

$$\Leftrightarrow \quad \alpha_t > 1. \tag{41}$$

However, if $\alpha_t > 1$, then $\hat{Q}_{S_t^{obs}}(1-\alpha_t) = -\infty \Rightarrow \mathrm{err}_t = 1$. This is because the quantile will be the trivially small negative infinite quantile, meaning there will definitely be miscoverage at $t$. Then,

$$\Rightarrow \alpha_{t+1} = \alpha_t + \frac{\gamma}{p_t}(\alpha - \mathrm{err}_t) = \alpha_t + \frac{\gamma}{p_t}(\alpha - 1) \leq \alpha_t \tag{42}$$

since $\frac{\gamma}{p_t}(\alpha - 1)$ is is negative by definition of $\alpha$. We have reached a contradiction with $\alpha_{t+1}$ being the first supremum value reached. $\square$

Lemma 2 will enable us to prove Proposition 2. The proof is derivative of the constant feedback ACI proof in (Gibbs & Candes, 2021), and the key idea is to bound the expectation of $\alpha_{t+1}$.

*Proof.* Examine the expectation of $\alpha_{t+1}$, conditional on $D_{T+1} := \{\mathrm{err}_t, p_t\}_{t\leq T+1}$:

$$\mathbb{E}_{\mathrm{obs}_1 \sim p_1, \ldots, \mathrm{obs}_t \sim p_{T+1}}[\alpha_{T+1}|D_{T+1}] = \alpha_1 + \gamma \sum_{t=1}^{T+1}(\alpha - \mathrm{err}_t) \tag{43}$$

We will abbreviate the left hand side by $\mathbb{E}[\alpha_{T+1}|D_{T+1}]$. Because the expected value cannot exceed the range of the value of $\alpha_{T+1}$, we infer that $\mathbb{E}[\alpha_{T+1}|D_{T+1}] \in [-\frac{\gamma}{M_{T+1}}, 1 + \frac{\gamma}{M_{T+1}}]$.

First, we observe from Lemma 2 that

$$\mathbb{E}[\alpha_{T+1}|D_{T+1}] = \alpha_1 + \sum_{t=1}^{T} \gamma(\alpha - \text{err}_t) \in [-\frac{\gamma}{M_{T+1}}, 1 + \frac{\gamma}{M_{T+1}}] \qquad (44)$$

$$\implies \quad -\sum_{t=1}^{T} \gamma(\alpha - \text{err}_t) = \alpha_1 - \mathbb{E}[\alpha_{T+1}|D_{T+1}] \qquad (45)$$

$$\implies \quad \left|\frac{1}{T}\sum_{t=1}^{T} \text{err}_t - \alpha\right| = \frac{|\alpha_1 - \mathbb{E}[\alpha_{T+1}|D_{T+1}]|}{T\gamma}. \qquad (46)$$

To bound the right hand side above, consider in turn the two cases of $\alpha_1 - \mathbb{E}[\alpha_{T+1}|D_{T+1}] \geq 0$ and $\alpha_1 - \mathbb{E}[\alpha_{T+1}|D_{T+1}] < 0$.

Starting with Case 1,

$$\alpha_1 - \mathbb{E}[\alpha_{T+1}|D_{T+1}] \geq 0 \Rightarrow \alpha_1 \geq \mathbb{E}[\alpha_{T+1}|D_{T+1}] \Rightarrow \alpha_{T+1} \in [-\frac{\gamma}{M_{T+1}}, \alpha_1]. \qquad (47)$$

This case corresponds to the following equivalence: $|\alpha_1 - \mathbb{E}[\alpha_{T+1}|D_{T+1}]| = \alpha_1 - \mathbb{E}[\alpha_{T+1}|D_{T+1}]$. Negating $\alpha_{T+1} \geq -\frac{\gamma}{M_{T+1}}$, we get $-\alpha_{T+1} \leq \frac{\gamma}{M_{T+1}}$. Thus,

$$|\alpha_1 - \mathbb{E}[\alpha_{T+1}|D_{T+1}]| = \alpha_1 - \mathbb{E}[\alpha_{T+1}|D_{T+1}] \leq \alpha_1 + \frac{\gamma}{M_{T+1}} \qquad (48)$$

$$\implies \quad \frac{|\alpha_1 - \mathbb{E}[\alpha_{T+1}|D_{T+1}]|}{T\gamma} \leq \frac{\alpha_1 + \frac{\gamma}{M_{T+1}}}{T\gamma}. \qquad (49)$$

In Case 2,

$$\alpha_1 - \mathbb{E}[\alpha_{T+1}|D_{T+1}] < 0 \Rightarrow \alpha_1 < \mathbb{E}[\alpha_{T+1}|D_{T+1}] \Rightarrow \mathbb{E}[\alpha_{T+1}|D_{T+1}] \in (\alpha_1, 1 + \frac{\gamma}{M_{T+1}}]. \qquad (50)$$

This case corresponds to the following equivalence: $|\alpha_1 - \mathbb{E}[\alpha_{T+1}|D_{T+1}]| = \mathbb{E}[\alpha_{T+1}|D_{T+1}] - \alpha_1 = -1(\alpha_1 - \mathbb{E}[\alpha_{T+1}|D_{T+1}])$. Plugging in $\mathbb{E}[\alpha_{T+1}|D_{T+1}] < 1 + \frac{\gamma}{M_{T+1}}$,

$$|\alpha_1 - \mathbb{E}[\alpha_{T+1}|D_{T+1}]| = \mathbb{E}[\alpha_{T+1}|D_{T+1}] - \alpha_1 \leq 1 + \frac{\gamma}{M_{T+1}} - \alpha_1 \qquad (51)$$

$$\frac{|\alpha_1 - \mathbb{E}[\alpha_{T+1}|D_{T+1}]|}{T\gamma} \leq \frac{(1 - \alpha_1) + \frac{\gamma}{M_{T+1}}}{T\gamma}. \qquad (52)$$

Lastly, we merge the two cases by taking the maximum over $\{\alpha_1, 1 - \alpha_1\}$ to come up with an upper bound that covers both cases.

$$\left|\frac{1}{T}\sum_{t=1}^{T} \text{err}_t - \alpha\right| = \frac{|\alpha_1 - \mathbb{E}[\alpha_{T+1}|D_{T+1}]|}{T\gamma} \leq \frac{\max\{\alpha_1, 1 - \alpha_1\} + \frac{\gamma}{M_{T+1}}}{T\gamma}. \qquad (53)$$

Taking the limit as $T \to \infty$, if $M_\infty$ is bounded, we get $\lim_{T\to\infty} \left|\frac{1}{T}\sum_{t=1}^{T} \text{err}_t - \alpha\right| = 0$, as claimed. $\qquad\square$

## E    EXTENDED EXPERIMENTS: INTERMITTENT QUANTILE TRACKING

We experiment with these four predictors because of their use in prior conformal literature (Angelopoulos et al., 2023) and in order to ensure that under different base model conditions, IQT maintains coverage close to the desired level. (1) Autoregressive (**AR**) model with 3 lags, (2) **Theta** model with $\theta = 2$ (Assimakopoulos & Nikolopoulos, 2000), (3) **Prophet** model Taylor & Letham (2018), and (4) **Transformer** model (Vaswani et al., 2017). Consistent with prior works, for all base models except for transformer, we retrain the base model after each timestep; for the transformer, we retrain every 100 timesteps. We set lookback window $k = 100$ timesteps for the Google and Amazon stock price data, and $k = 300$ for the Elec2 dataset.

Table 1: **IQT on Amazon Stocks (*infrequent observations*): Trends Across Models.** We test four base models, set $p_t = 0.1, \forall t$ to simulate seeing the true price only 10% of the time, and report the mean across 5 seeds.

| Metric | lr | AR Base Model | | Prophet Base Model | | Theta Base Model | | Transformer Base Model | |
|---|---|---|---|---|---|---|---|---|---|
| | | IQT-pi | IQT-pd | IQT-pi | IQT-pd | IQT-pi | IQT-pd | IQT-pi | IQT-pd |
| Coverage | 1.0 | 0.903 | 0.937 | 0.906 | 0.935 | 0.899 | 0.921 | 0.902 | 0.923 |
| | 0.1 | 0.876 | 0.903 | 0.876 | 0.906 | 0.815 | 0.899 | 0.808 | 0.902 |
| | 0.01 | 0.690 | 0.876 | 0.690 | 0.876 | 0.430 | 0.815 | 0.360 | 0.808 |
| Longest err seq | 1.0 | 13.2 | 13.2 | 11.6 | 30.6 | 27 | 41.2 | 28.8 | 30 |
| | 0.1 | 6.8 | 20.8 | 10 | 11.6 | 51.8 | 27 | 89 | 28.8 |
| | 0.01 | 12.6 | 6.8 | 13.8 | 10 | 145.2 | 51.8 | 212.8 | 89 |
| Avg set size | 1.0 | 47.452 | 425.435 | 46.056 | 414.977 | 105.040 | 711.829 | 131.025 | 982.469 |
| | 0.1 | 16.872 | 47.452 | 16.495 | 46.056 | 82.463 | 105.040 | 118.040 | 131.025 |
| | 0.01 | 9.428 | 16.872 | 9.470 | 16.495 | 35.458 | 82.463 | 46.929 | 118.040 |

Table 2: **IQT on Amazon Stocks (*partial observations*): Trends Across Models.** We test four base models, set $p_t = 0.5, \forall t$ to simulate seeing the true price only 50% of the time, and report the mean across 5 seeds.

| Metric | lr | AR | | Prophet | | Theta | | Transformer | |
|---|---|---|---|---|---|---|---|---|---|
| | | IQT-pi | IQT-pd | IQT-pi | IQT-pd | IQT-pi | IQT-pd | IQT-pi | IQT-pd |
| Marginal coverage | 1.0 | 0.902 | 0.905 | 0.901 | 0.904 | 0.906 | 0.904 | 0.900 | 0.904 |
| | 0.1 | 0.891 | 0.896 | 0.883 | 0.893 | 0.890 | 0.895 | 0.879 | 0.885 |
| | 0.01 | 0.846 | 0.869 | 0.737 | 0.812 | 0.843 | 0.870 | 0.705 | 0.796 |
| Longest err seq | 1.0 | 5.6 | 5.8 | 7.4 | 8 | 5.6 | 6.6 | 7.8 | 8 |
| | 0.1 | 4.6 | 4 | 12.8 | 8.2 | 7.6 | 5 | 23.4 | 11.4 |
| | 0.01 | 7.2 | 7 | 89.4 | 50.6 | 10 | 10 | 191.2 | 97.6 |
| Avg set size | 1.0 | 43.816 | 78.447 | 76.013 | 139.875 | 44.449 | 77.741 | 100.127 | 183.529 |
| | 0.1 | 17.535 | 19.918 | 66.940 | 55.233 | 17.581 | 19.854 | 92.513 | 72.867 |
| | 0.01 | 13.684 | 14.952 | 70.972 | 81.854 | 13.678 | 15.085 | 99.109 | 118.457 |

**Amazon stock price data under partial ($p_t = 0.5$) feedback.** Figure 8 shows the prediction interval sizes for 1 seed, and coverage averaged over 5 seeds for Amazon stock price data under partial ($p_t = 0.5$) feedback. We see that the interval size for **IQT-pd** is larger than **IQT-pi** for high learning rate lr $= 1$, but the size of intervals is comparable for smaller learning rates. Table 2 shows the performance metrics averaged over 5 seeds.

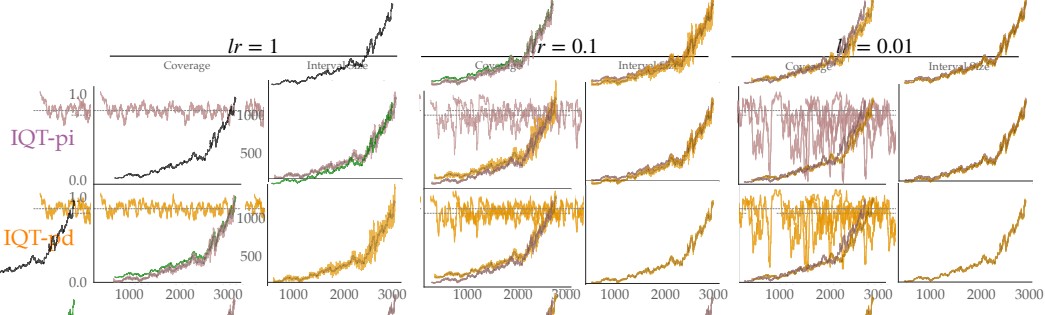

Figure 8: Amazon stock price data under partial ($p_t = 0.5$) feedback. We show the prediction interval sizes for 1 seed, and coverage averaged over 5 seeds for Amazon stock price data under partial observations.

**Amazon stock price data under partial ($p_t = 0.9$) feedback.** Figure 9 shows the prediction interval sizes for 1 seed, and coverage averaged over 5 seeds for Amazon stock price data under frequent ($p_t = 0.9$) feedback. We see that the interval size for **IQT-pd** and **IQT-pi** are very similar across learning rates. Table 3 shows the performance metrics averaged over 5 seeds.

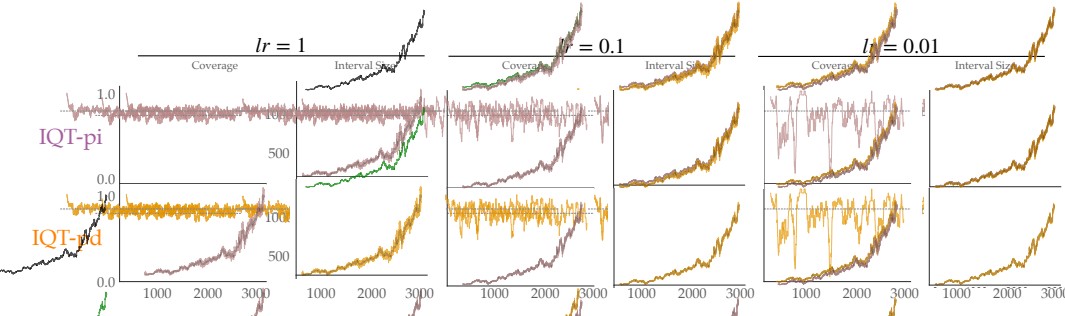

Figure 9: Amazon stock price data under frequent ($p_t = 0.9$) feedback. We show the prediction interval sizes for 1 seed, and coverage averaged over 5 seeds for Amazon stock price data under partial observations.

Table 3: **IQT on Amazon Stocks (*frequent observations*): Trends Across Models.** We test four base models, set $p_t = 0.9, \forall t$ to simulate seeing the true price 90% of the time, and report the mean across 5 seeds.

| Metric | lr | AR | | Prophet | | Theta | | Transformer | |
|---|---|---|---|---|---|---|---|---|---|
| | | IQT-pi | IQT-pd | IQT-pi | IQT-pd | IQT-pi | IQT-pd | IQT-pi | IQT-pd |
| Marginal coverage | 1.0 | 0.902 | 0.903 | 0.901 | 0.903 | 0.902 | 0.902 | 0.903 | 0.902 |
| | 0.1 | 0.894 | 0.894 | 0.886 | 0.886 | 0.8932 | 0.894 | 0.885 | 0.887 |
| | 0.01 | 0.870 | 0.873 | 0.797 | 0.805 | 0.871 | 0.873 | 0.786 | 0.798 |
| Longest err seq | 1.0 | 2.8 | 2.6 | 3 | 3.6 | 3.2 | 3.6 | 3.2 | 3 |
| | 0.1 | 3.6 | 3.8 | 7 | 6.6 | 5.4 | 5 | 10.6 | 9 |
| | 0.01 | 7 | 7 | 57.8 | 44.2 | 10 | 10 | 97.8 | 98.4 |
| Avg set size | 1.0 | 43.221 | 46.624 | 71.646 | 78.889 | 42.979 | 46.270 | 93.068 | 103.777 |
| | 0.1 | 17.663 | 17.982 | 54.235 | 52.265 | 17.687 | 18.121 | 69.700 | 66.835 |
| | 0.01 | 14.627 | 14.832 | 79.736 | 81.078 | 14.732 | 14.954 | 116.272 | 117.762 |

Table 4: **IQT on Google Stocks (*infrequent observations*): Trends Across Models.** We test four base models, set $p_t = 0.1, \forall t$ to simulate seeing the true price only 10% of the time, and report the mean across 5 seeds.

| Metric | lr | AR | | Prophet | | Theta | | Transformer | |
|---|---|---|---|---|---|---|---|---|---|
| | | IQT-pi | IQT-pd | IQT-pi | IQT-pd | IQT-pi | IQT-pd | IQT-pi | IQT-pd |
| Marginal coverage | 1.0 | 0.925 | 0.949 | 0.928 | 0.929 | 0.929 | 0.929 | 0.907 | 0.927 |
| | 0.1 | 0.910 | 0.925 | 0.897 | 0.928 | 0.906 | 0.929 | 0.793 | 0.907 |
| | 0.01 | 0.812 | 0.910 | 0.639 | 0.897 | 0.692 | 0.906 | 0.234 | 0.793 |
| Longest err seq | 1.0 | 11.4 | 27.8 | 21.4 | 42.4 | 17.2 | 29.2 | 23.2 | 29 |
| | 0.1 | 6 | 11.4 | 35.6 | 21.4 | 53.4 | 17.2 | 109.2 | 23.2 |
| | 0.01 | 9.2 | 6 | 69.6 | 35.6 | 103 | 53.4 | 873.4 | 109.2 |
| Avg set size | 1.0 | 87.834 | 911.476 | 134.392 | 1017.708 | 93.068 | 815.183 | 218.270 | 1445.050 |
| | 0.1 | 24.305 | 87.834 | 104.224 | 134.392 | 49.364 | 93.068 | 269.950 | 218.270 |
| | 0.01 | 15.070 | 24.305 | 54.653 | 104.224 | 28.910 | 49.364 | 109.453 | 269.950 |

### E.1 GOOGLE STOCK PRICE DATASET RESULTS

For the Google stock price dataset, Tables 4, 5, and 6 show the results for infrequent, partial, and frequent feedback. Under infrequent observations, **IQT-pd** boosts coverage over **IQT-pi**, at the cost of larger prediction intervals on average. As the observation probability increases, the discrepancy in prediction interval size between **IQT-pd** and **IQT-pi** narrows, and the coverage is very similar. At a high level, the trends in performance for **IQT-pd** and **IQT-pi** across the three observation levels are very similar for the Google dataset and the Amazon stock price dataset.

### E.2 ELEC2 DATASET RESULTS

For the Elec2 dataset, Tables 7, 8, and 9 show the results for infrequent, partial, and frequent feedback. Similar to the other two datasets, under infrequent observations, **IQT-pd** boosts coverage over **IQT-pi**, at the cost of larger prediction intervals on average. As the observation probability increases, the discrepancy in prediction interval size between **IQT-pd** and **IQT-pi** narrows, and the coverage is very

Table 5: **IQT on Google Stocks (*partial observations*): Trends Across Models.** We test four base models, set $p_t = 0.5, \forall t$ to simulate seeing the true price only 50% of the time, and report the mean across 5 seeds.

| Metric | lr | AR | | Prophet | | Theta | | Transformer | |
|---|---|---|---|---|---|---|---|---|---|
| | | IQT-pi | IQT-pd | IQT-pi | IQT-pd | IQT-pi | IQT-pd | IQT-pi | IQT-pd |
| Marginal coverage | 1.0 | 0.909 | 0.909 | 0.902 | 0.908 | 0.906 | 0.903 | 0.902 | 0.899 |
| | 0.1 | 0.905 | 0.905 | 0.896 | 0.892 | 0.899 | 0.901 | 0.901 | 0.894 |
| | 0.01 | 0.891 | 0.901 | 0.858 | 0.887 | 0.864 | 0.890 | 0.661 | 0.809 |
| Longest err seq | 1.0 | 5.2 | 6.4 | 8 | 7.6 | 5.4 | 7.8 | 8.2 | 8.4 |
| | 0.1 | 4 | 4 | 13.8 | 11 | 13.8 | 7.4 | 16 | 13.2 |
| | 0.01 | 6 | 6 | 49.4 | 32.2 | 80.8 | 55.6 | 239.8 | 104.2 |
| Avg set size | 1.0 | 59.236 | 101.424 | 91.522 | 168.522 | 64.736 | 116.381 | 131.415 | 241.408 |
| | 0.1 | 22.021 | 25.670 | 71.972 | 63.493 | 36.04 | 34.893 | 156.157 | 106.766 |
| | 0.01 | 18.820 | 19.830 | 91.394 | 96.245 | 42.575 | 43.364 | 246.053 | 276.761 |

Table 6: **IQT on Google Stocks (*frequent observations*): Trends Across Models.** We test four base models, set $p_t = 0.9, \forall t$ to simulate seeing the true price 90% of the time, and report the mean across 5 seeds.

| Metric | lr | AR | | Prophet | | Theta | | Transformer | |
|---|---|---|---|---|---|---|---|---|---|
| | | IQT-pi | IQT-pd | IQT-pi | IQT-pd | IQT-pi | IQT-pd | IQT-pi | IQT-pd |
| Marginal coverage | 1.0 | 0.903 | 0.902 | 0.902 | 0.903 | 0.902 | 0.902 | 0.902 | 0.901 |
| | 0.1 | 0.897 | 0.898 | 0.889 | 0.889 | 0.898 | 0.898 | 0.892 | 0.892 |
| | 0.01 | 0.895 | 0.897 | 0.880 | 0.881 | 0.889 | 0.891 | 0.791 | 0.808 |
| Longest err seq | 1.0 | 2.8 | 3 | 3.2 | 2.8 | 3.6 | 2.8 | 3 | 3.6 |
| | 0.1 | 4 | 4 | 8 | 7.8 | 6.8 | 6.6 | 10 | 9.4 |
| | 0.01 | 6 | 6 | 32.4 | 31.4 | 58 | 57.6 | 108.4 | 105 |
| Avg set size | 1.0 | 53.161 | 57.376 | 85.427 | 93.799 | 61.876 | 68.177 | 121.994 | 134.782 |
| | 0.1 | 21.068 | 21.564 | 59.732 | 57.680 | 31.262 | 31.240 | 106.240 | 99.712 |
| | 0.01 | 19.096 | 19.396 | 94.682 | 94.370 | 43.206 | 42.881 | 278.926 | 279.124 |

Table 7: **IQT on Elec2 Dataset (*infrequent observations*): Trends Across Models.** We test four base models, set $p_t = 0.1, \forall t$ to simulate seeing the true price only 10% of the time, and report the mean across 5 seeds.

| Metric | lr | AR | | Prophet | | Theta | | Transformer | |
|---|---|---|---|---|---|---|---|---|---|
| | | IQT-pi | IQT-pd | IQT-pi | IQT-pd | IQT-pi | IQT-pd | IQT-pi | IQT-pd |
| Marginal coverage | 1.0 | 0.918 | 0.908 | 0.900 | 0.934 | 0.909 | 0.949 | 0.903 | 0.904 |
| | 0.1 | 0.897 | 0.918 | 0.901 | 0.900 | 0.911 | 0.909 | 0.908 | 0.903 |
| | 0.01 | 0.809 | 0.897 | 0.665 | 0.901 | 0.857 | 0.911 | 0.663 | 0.908 |
| Longest err seq | 1.0 | 10.4 | 24.8 | 19.6 | 28.6 | 15.2 | 24.4 | 20.2 | 25.4 |
| | 0.1 | 4 | 10.4 | 19.2 | 19.6 | 9.2 | 15.2 | 10.8 | 20.2 |
| | 0.01 | 5.2 | 4 | 29.8 | 19.2 | 13 | 9.2 | 29.4 | 10.8 |
| Avg set size | 1.0 | 0.120 | 1.867 | 0.846 | 6.374 | 0.252 | 2.489 | 0.905 | 6.565 |
| | 0.1 | 0.091 | 0.120 | 0.528 | 0.846 | 0.085 | 0.252 | 0.556 | 0.905 |
| | 0.01 | 0.061 | 0.091 | 0.307 | 0.528 | 0.059 | 0.085 | 0.349 | 0.556 |

similar. The trends in performance for **IQT-pd** and **IQT-pi** across the three observation levels are very similar for the Elec2 dataset and the Amazon stock price dataset.

# F FURTHER EXAMINATION: SIMULATED EXPERIMENTS

**On Expert Realizability.** An outstanding challenge in the theory of DAgger is guaranteeing learning a high-quality policy from the expert when the expert is unrealizable. We assume, much like Spencer et al. (2021), that our expert is realizable. Whether or not this is true in practice with human experts is an open area of research. Nevertheless, similar to HG-DAgger Kelly et al. (2019) and EnsembleDAgger Menda et al. (2019), our approach offers a practical approach to contend with these real-world challenges.

Table 8: **IQT on Elec2 Dataset (*partial observations*): Trends Across Models.** We test four base models, set $p_t = 0.5, \forall t$ to simulate seeing the true price only 50% of the time, and report the mean across 5 seeds.

| Metric | lr | AR | | Prophet | | Theta | | Transformer | |
|---|---|---|---|---|---|---|---|---|---|
| | | IQT-pi | IQT-pd | IQT-pi | IQT-pd | IQT-pi | IQT-pd | IQT-pi | IQT-pd |
| Marginal coverage | 1.0 | 0.909 | 0.905 | 0.898 | 0.908 | 0.907 | 0.909 | 0.902 | 0.899 |
| | 0.1 | 0.895 | 0.896 | 0.901 | 0.896 | 0.901 | 0.899 | 0.903 | 0.897 |
| | 0.01 | 0.892 | 0.894 | 0.865 | 0.887 | 0.899 | 0.901 | 0.886 | 0.903 |
| Longest err seq | 1.0 | 4.6 | 5.2 | 8.6 | 7.4 | 5.2 | 6.4 | 6.8 | 7.6 |
| | 0.1 | 3.8 | 5 | 7.2 | 7.4 | 5.2 | 4.8 | 7.8 | 7.6 |
| | 0.01 | 3.2 | 2.8 | 27 | 14.2 | 4.2 | 4.6 | 11 | 9.8 |
| Avg set size | 1.0 | 0.187 | 0.325 | 0.721 | 1.304 | 0.234 | 0.443 | 0.738 | 1.287 |
| | 0.1 | 0.093 | 0.101 | 0.475 | 0.4721 | 0.074 | 0.087 | 0.487 | 0.478 |
| | 0.01 | 0.087 | 0.088 | 0.481 | 0.500 | 0.068 | 0.070 | 0.497 | 0.517 |

Table 9: **IQT on Elec2 Dataset (*frequent observations*): Trends Across Models.** We test four base models, set $p_t = 0.9, \forall t$ to simulate seeing the true price 90% of the time, and report the mean across 5 seeds.

| Metric | lr | AR | | Prophet | | Theta | | Transformer | |
|---|---|---|---|---|---|---|---|---|---|
| | | IQT-pi | IQT-pd | IQT-pi | IQT-pd | IQT-pi | IQT-pd | IQT-pi | IQT-pd |
| Marginal coverage | 1.0 | 0.902 | 0.901 | 0.901 | 0.901 | 0.902 | 0.902 | 0.902 | 0.902 |
| | 0.1 | 0.900 | 0.902 | 0.897 | 0.898 | 0.897 | 0.899 | 0.897 | 0.898 |
| | 0.01 | 0.903 | 0.903 | 0.881 | 0.883 | 0.893 | 0.894 | 0.899 | 0.902 |
| Longest err seq | 1.0 | 2.6 | 3 | 3.8 | 3.4 | 2.6 | 2.8 | 3.4 | 3.2 |
| | 0.1 | 2.6 | 2.8 | 4.6 | 4.8 | 4.2 | 3.8 | 5.8 | 5.2 |
| | 0.01 | 2.4 | 2.8 | 15.6 | 15 | 4.6 | 4.8 | 10 | 10 |
| Avg set size | 1.0 | 0.179 | 0.192 | 0.677 | 0.734 | 0.219 | 0.243 | 0.670 | 0.724 |
| | 0.1 | 0.096 | 0.0977 | 0.446 | 0.437 | 0.070 | 0.071 | 0.444 | 0.438 |
| | 0.01 | 0.091 | 0.092 | 0.492 | 0.492 | 0.066 | 0.067 | 0.513 | 0.518 |

**Implementation Details.** Both **EnsembleDAgger** and **ConformalDAgger** are implemented as 7-layer multilayer perceptions (with hidden sizes [64,128,472,512,256,64,42]). We use ReLU activations between each layer. We train using a learning rate of 0.001 and a batch size of 32. We train the initial policy for 200 iterations, and fine-tune between each interactive deployment episode for 100 iterations. The **EnsembleDAgger** safety classifier is implemented as a 4-layer perception ([64,128,64,42]) with ReLU activation between each layer and we apply a sigmoid function on the output to classify the output.

**A closer look at expert shift under $p_t = 0.2$ intermittent feedback .** When the expert shift occurs at episode 5 (Figure 10), the human provides feedback occasionally to the learners which deviates from the predicted actions. **ConformalDAgger** increases its calibrated uncertainty based on these human inputs, and once uncertainty exceeds the threshold, the probability for asking for help converges to 1, causing the robot to ask for more human feedback such that it can learn the new goal.

**Simulated results under partial ($p_t = 0.5$) intermittent feedback .** Under $p_t = 0.5$ intermittent feedback (Figure 11), **ConformalDAgger** is able to detect the shift and drift more quickly than **EnsembleDAgger**. We find that as the **ConformalDAgger** algorithm under the stationary expert becomes more noisy has increased miscoverage. This is due to the expert labels decreasing the conformal parameters, $q^{hi}, q^{lo}$ as the expert gives feedback, causing the intervals to become too small. Under the short time horizon, the miscoverage rate is higher than the desired level.

**Simulated results under frequent ($p_t = 0.9$) intermittent feedback .** Under $p_t = 0.9$ intermittent feedback (Figure 12), both algorithms are able to quickly adapt to expert shift and drift. This is because both algorithms during deployment are receiving human labels extremely frequently. Similar to partial feedback, as the **ConformalDAgger** algorithm under the stationary expert becomes more noisy has increased miscoverage. **ConformalDAgger** decreases the value of the conformal parameters, $q^{hi}, q^{lo}$ as the expert gives feedback, causing the intervals to become too small, giving miscoverage higher than the desired level.

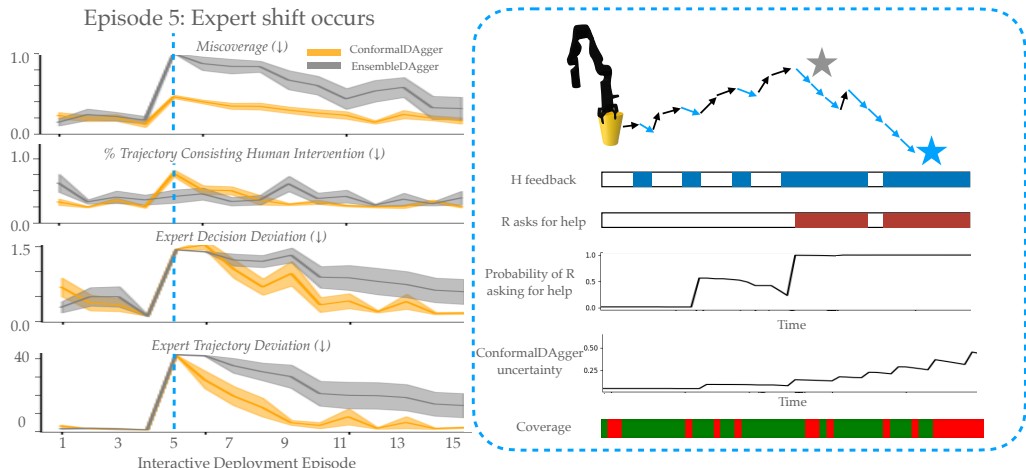

Figure 10: At episode 5, the **ConformalDAgger** learner uncertainty enables the robot to ask for more human feedback to gather information about the shifted goal.

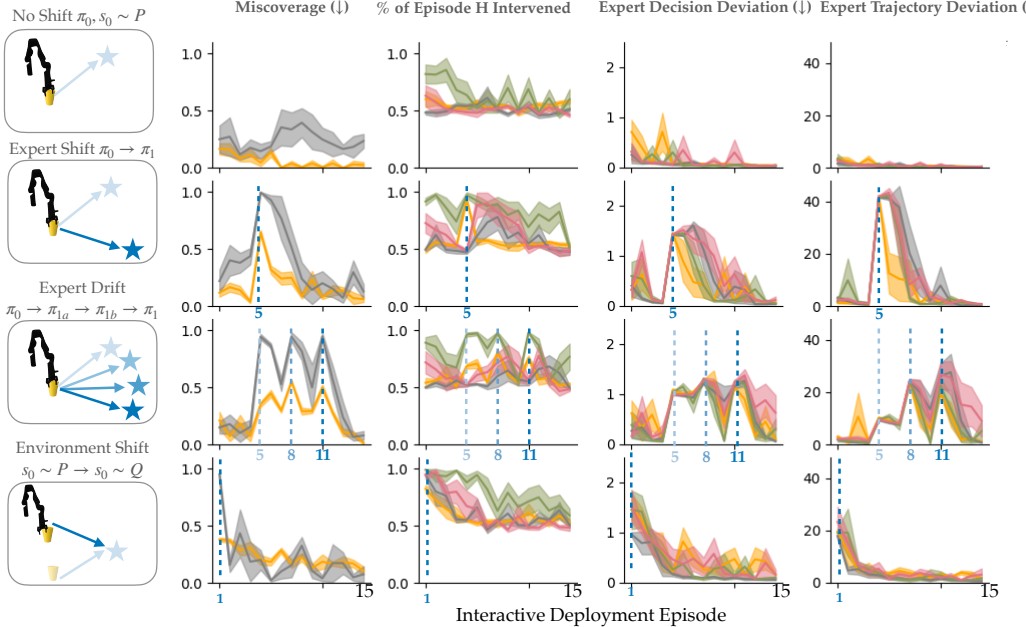

Figure 11: Under $p_t = 0.5$ intermittent feedback, **ConformalDAgger** learns the shift and drift more quickly than **EnsembleDAgger** and **SafeDAgger**, with fewer queries than **LazyDAgger**.

## G  EXPERIMENTS IN ROBOSUITE NUT-ASSEMBLY BENCHMARK

We conducted additional experiments on Robosuite's nut-assembly task, a benchmark for LazyDAgger Hoque et al. (2021b) and ThriftyDAgger Hoque et al. (2021a). For consistency, we used the LazyDAgger thresholds from Hoque et al. (2021a) and set the SafeDAgger threshold to match the safety value used in LazyDAgger. The nut-assembly task involved a hardcoded expert policy that sequentially rotates to pick up the nut, lifts it, moves it to the peg, and lowers it to place. We trained the base policy with 30 demonstration episodes.

We evaluated this task under two conditions: a No-Shift scenario, where the task remains unchanged during deployment, and an Environment-Shift scenario, where the peg location is altered during interactive deployment (highlighted in red in Figure 13).

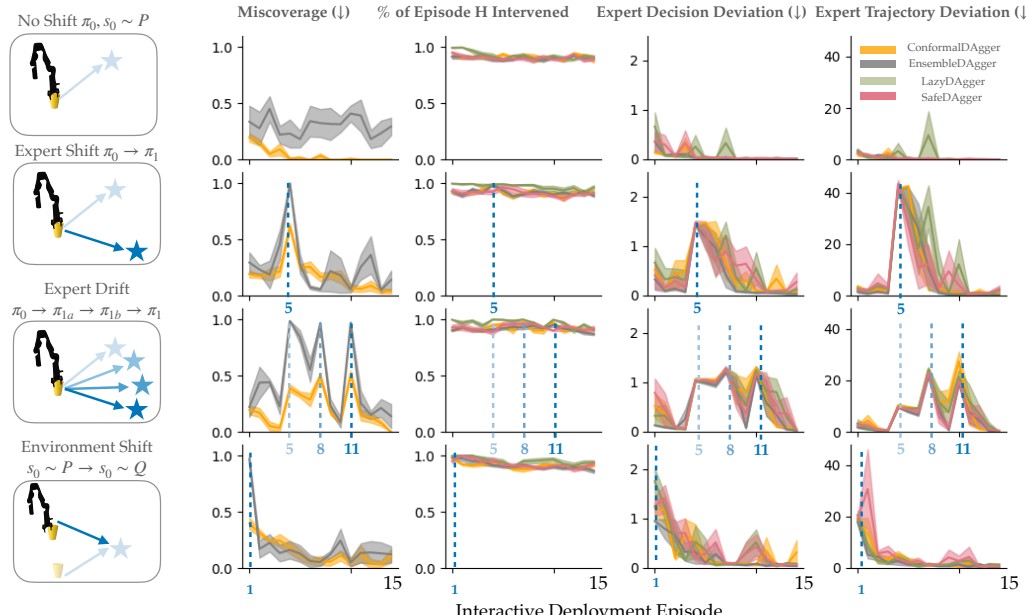

Figure 12: Under $p_t = 0.9$ intermittent feedback, all approaches learn the shift and drift quickly.

Performance metrics included: (1) autonomous success rate—success rate in deployment without a human supervisor (50 rollouts), and (2) intervention-aided success rate—success rate with a human supervisor in the loop. During training, we also tracked the number of interventions, human actions, and robot actions per episode. These metrics were calculated only for successful episodes to avoid bias from maximum episode length, which can inflate action counts for less successful policies that frequently reach the time limit.

The task required grasping a ring in a random initial pose and threading it onto a cylinder at a fixed target location. This involved two key challenges motivating learning from interventions: (1) successfully grasping the ring, and (2) accurately placing it over the cylinder (Figure 13). A simulated human provided interventions by teleoperating the robot. The state inputs included the robot's joint angles and the ring's pose, while the actions involved 3D translation, 3D rotation, and gripper control. We used 30 publicly available offline demonstrations (https://github.com/ryanhoque/thriftydagger 2,687 state-action pairs) from a human supervisor to initialize the robot policy across all algorithms.

ConformalDAgger identified the initial policy's nonperformance in both contexts, prompting more queries and achieving a higher autonomous rollout success rate compared to other baselines.

# H   IMPLEMENTATION AND TASK DETAILS: REAL EXPERIMENTS

## H.1   LEARNING POLICY TRAINING DETAILS

We record robot and human actions at 15hz. The initial policy $\pi_0^r$ is trained for 60K iterations with a batch size of 100. We use a weight decay of 1.0e-6, learning rate of 0.0001; for learning-rate scheduling, we used a cosine schedule with linear warmup (Nichol & Dhariwal, 2021). The number of training diffusion iterations is 100, and number of inference diffusion iterations is 16. The policy is trained on an NVIDIA RTX A6000 GPU.

**A look at the real-world feedback request interface.** Figure 14 shows the real-world interface for requesting help from the user during the interactive deployment episodes. The user provides teleoperated actions via a Meta Quest 3, and the robot's uncertainty is displayed on a computer screen next to the robot. When the robot needs help, the robot pauses its execution and presents an alert notification on the screen.

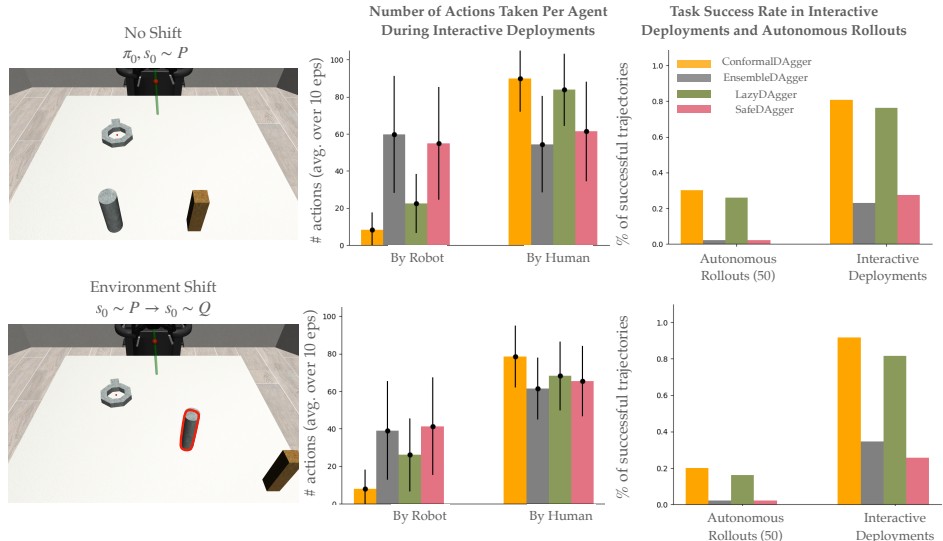

Figure 13: ConformalDAgger recognizes the learned policy from 30 demos is still nonperformant in both contexts, and increases the number of questions asked, resulting in a higher autonomous rollout success rate than other baselines.

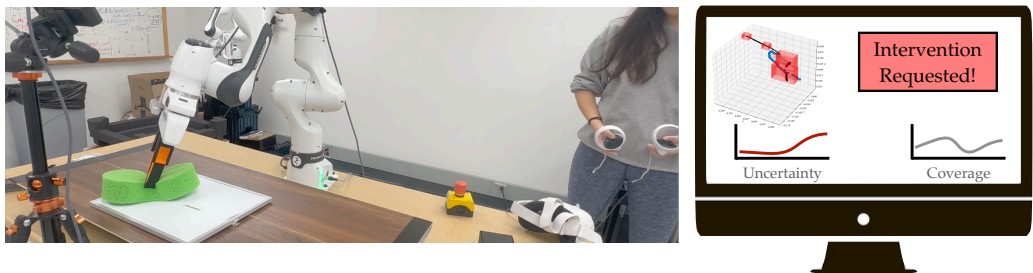

Figure 14: When the robot needs help, the robot pauses its execution and presents an alert notification on the screen.

**Sponging Task Specifications.** We present the task setup details (Figure 15) so that readers can also reproduce this task.

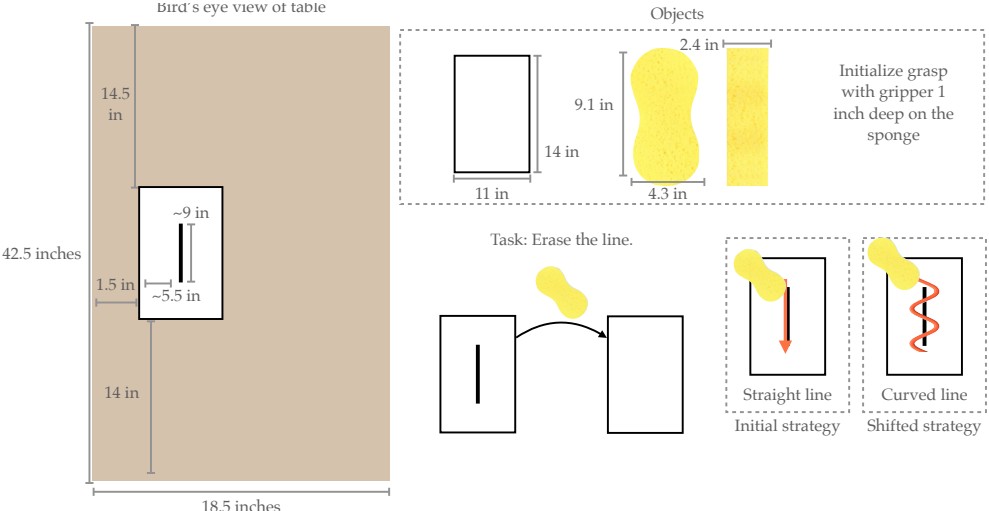

Figure 15: Sponge task specifications.