# OpenReview forum: "Conformalized Interactive Imitation Learning: Handling Expert Shift and Intermittent Feedback"
_ICLR.cc/2025/Conference — ICLR 2025 Poster_

### Official Review · Reviewer_jhY9 · 2024-10-29

**Soundness:** 3
**Presentation:** 3
**Contribution:** 2
**Rating:** 6
**Confidence:** 4

**Summary:**

This paper presents an interactive imitation learning method (ConformalDAgger) that has the learner (robot) actively ask for additional supervision when it is uncertain during deployment. As in prior methods, it uses uncertainty quantification to do this, but unlike other methods, it updates the uncertainty quantification online using the expert feedback during deployment. The paper does this by introducing a novel online conformal prediction method called IQT, which is able to deal with intermittent expert labels, unlike prior approaches in online conformal prediction. The proposed method is shown to be effective at dealing with situations where there is Expert Shift, i.e. the expert policy changes between train and test time.

**Strengths:**

The paper is well-written and mostly easy to follow. The methods presented in the paper are interesting and appear to be novel. There is empirical evidence showing the ability of the proposed method to deal with "Expert Shift" in both simulated and real-world experiments.

**Weaknesses:**

**Weak and Unclear Motivation.** The proposed method seems to be exclusively useful for the setting of "Expert Shift" where the expert policy changes between train-time and test-time. This does not seem to be a very practical, commonplace issue. Are there other motivating examples for the proposed method as well, and corresponding settings where the proposed method would outperform other baselines? This aligns with Takeaway 3 in Section 5 too, where performance is similar to the baseline in the absence of expert shift. On a related note, a change in the expert policy can be arbitrary - it would be useful to provide some more intuition on what the best behavior should be for an ideal method in sensing and dealing with such a shift.

**Thoroughness and Reproducibility of Experiments.** The paper could benefit from some more thorough simulation experiments. For example, including a simulation experiment with high-dimensional observations (e.g. images) similar to the real-world experiments could be beneficial. Also running such experiments on some standard benchmark tasks in existing simulation frameworks that go beyond goal-reaching would greatly improve the reproducibility of the work and the experiments. Some suggestions include [robomimic](https://robomimic.github.io/), [RLBench](https://sites.google.com/view/rlbench), [ManiSkill](https://www.maniskill.ai/home), or some of the tasks presented in [Diffusion Policy](https://diffusion-policy.cs.columbia.edu/). Also, the paper only compares against EnsembleDAgger but it seems like there are other suitable baselines that could also be appropriate, such as [ThriftyDAgger](https://sites.google.com/view/thrifty-dagger/home).

**Other Issues.**

- Algorithm 1, line 11 is potentially misleading. The expert is only selectively annotating states unlike DAgger where all states observed by the agent would be relabeled.
- several undefined Sec refs (line 154, line 212, line 252, line 471)

**Questions:**

- IQT is only applied to the end effector position (bottom of page 9) -- is this a limitation? There might be robot manipulation tasks that require more rotation-heavy actions.
- In the hardware robot experiment, why is the EnsembleDAgger baseline different from the one used in the simulation experiments?

---

> ### Author Response · Authors · 2024-11-23
> **Response to Reviewer jhY9: On Expert Shift Motivation**
>
> We thank the reviewer for the thoughtful and constructive feedback. With respect to weaknesses and questions, we provide our responses as follows. We have uploaded a revised manuscript with additional experiments, analysis, and baselines; we discuss the results below and refer the reader to the updated figures in the manuscript. We use the color **blue** to highlight updated text and figures in the manuscript.
> \
> &nbsp;
> ## A. Regarding expert shift motivation
> We would first like to clarify – algorithmically, our IQT approach is agnostic to the underlying cause of distribution shift between the expert and the policy. All that it requires is accessing the expert’s labels intermittently to construct calibrated intervals on the base policy’s predictions. The reason the base policy is inconsistent with the expert labels can come from a variety of underlying causal factors. In our initial submission, we focused on expert shift – wherein the expert’s latent strategy changes over times – as one concrete cause of why the base policy disagrees with the expert.
>
> More broadly, policy mispredictions during deployment may result from a multitude of reasons, including suboptimal or multimodal demonstrations, covariate shift (e.g., encountering new observations of the environment during policy execution), or expert shift. To show the generality of our IQT algorithm, we added a **Environment Shift** context, where the state distribution during deployment differs from training. We instantiated **Environment Shift** in both the toy 4D reaching task and the newly-added **Robosuite benchmark Nut-Assembly task** (Robosuite benchmark results can be found in Appendix Section G in revised manuscript).

---

> ### Author Response · Authors · 2024-11-23
> **Regarding Inclusion of Additional Baselines**
>
> ## B. Regarding additional baselines
> We appreciate the suggestion of including additional interactive imitation learning baselines! In our original manuscript, we focused on comparing ConformalDAgger with EnsembleDAgger because the ensemble output variance provides a measurable interval for coverage, enabling direct comparison. As suggested by the reviewer, we included two relevant DAgger approaches: **LazyDAgger**[2] and **SafeDAgger** [3]. SafeDAgger relies on switching to human control when the predicted action discrepancy exceeds a threshold, while LazyDAgger initiates human intervention based on a safety classifier and returns to autonomous control only when the discrepancy falls below an additional context-switching-threshold [2]. Similar to Ensembles, Monte-Carlo Dropout approaches (e.g. UAIL [1]) can also be used to estimate epistemic uncertainty in deep networks; we chose EnsembleDAgger because of prior works empirically observed by Lakshminarayanan et al. [2017], Lee et al. [2015] that training ensemble networks with random initialization works well in practice for capturing this epistemic uncertainty. We refrained from adding ThriftyDAgger because the algorithm objective is mitigating human feedback to be below some budget. Conformal, Ensemble, Lazy, and Safe do not budget, or cap, the number of human interventions received, but rather focus on handling feedback through any human intervention necessary.
>
> We have rerun experiments including these additional baselines and under broader environment shift conditions for the 4D Simulated Reaching Task. The results are discussed in the section below and figures are in the attached PDF (Figure 5 in Section 5.2 in revised manuscript).
> \
> &nbsp;
> ### Updated Results on Simulated 4DoF Reaching Task with Human Gated Probability (=0.2 and =0.5):
> When the expert shifts and drifts, ConformalDAgger asks for more help immediately compared to EnsembleDAgger and SafeDAgger, enabling the algorithm to more quickly align to the expert, with fewer human interventions than LazyDAgger. LazyDAgger does successful align to the shifted strategy, but requests many more interventions overall.
>
> Under environment shift, EnsembleDAgger is well-calibrated, boosting the number of interventions requested. This is intuitive as ensembles capture the variance in the training distribution and when the environment shifts, changing the training distribution, the algorithm flags high uncertainty. ConformalDAgger performs well on environment shift context, though asks fewer questions than EnsembleDAgger, converging to the shifted policy nearly as quickly as EnsembleDAgger.
>
> When the human-gated probability is high (0.9), as expected, all approaches receive high quantities of human feedback, resulting in rapid converge to expert and environment shift.
>
> [1] Y. Cui, D. Isele, S. Niekum and K. Fujimura, "Uncertainty-Aware Data Aggregation for Deep Imitation Learning," 2019 International Conference on Robotics and Automation (ICRA), Montreal, QC, Canada, 2019, pp. 761-767, doi: 10.1109/ICRA.2019.8794025.
>
> [2] Hoque, Ryan, et al. "Lazydagger: Reducing context switching in interactive imitation learning." 2021 IEEE 17th international conference on automation science and engineering (case). IEEE, 2021.
>
> [3] Zhang, Jiakai, and Kyunghyun Cho. "Query-efficient imitation learning for end-to-end autonomous driving." arXiv preprint arXiv:1605.06450 (2016).

---

> ### Author Response · Authors · 2024-11-23
> **Regarding Experimental on Additional Simulated Benchmarks**
>
> ## On the inclusion of experiments on simulated IL benchmarks (e.g. Robosuite)
> We conducted additional experiments on Robosuite’s nut-assembly task, a benchmark for LazyDAgger and ThriftyDAgger. For consistency, we used the LazyDAgger thresholds from [1] and set the SafeDAgger threshold to match the safety value used in LazyDAgger. The nut-assembly task involved a hardcoded expert policy that sequentially rotates to pick up the nut, lifts it, moves it to the peg, and lowers it to place. We trained the base policy with 30 demonstration episodes.
>
> We evaluated this task under two conditions: a No-Shift scenario, where the task remains unchanged during deployment, and an Environment-Shift scenario, where the peg location is altered during interactive deployment (highlighted in red in Figure \ref{fig:robomimic}).
>
> A simulated human provided interventions by teleoperating the robot. We used 30 publicly available offline demonstrations (l{https://github.com/ryanhoque/thriftydagger} 2,687 state-action pairs) from a human supervisor to initialize the robot policy across all algorithms.
>
> We found that ConformalDAgger identified the initial policy's nonperformance in both contexts, prompting more queries and achieving a marginally higher autonomous rollout success rate compared to other baselines. See Figure 13 in Appendix Section G for updated results and figure.
>
> [1] Hoque, Ryan, et al. "Thriftydagger: Budget-aware novelty and risk gating for interactive imitation learning." (2021).

---

> ### Author Response · Authors · 2024-11-23
> **Regarding Questions and Minor Comments**
>
> - The expert is only selectively annotating states where the learner requests feedback. This is consistent with specifically online interactive imitation learning methods where the expert is in-the-loop during deployment rollouts.
>
> - We have fixed the Section references.
>
> - We applied IQT only to the end effector position because the Euclidean distance metric offers us an intuitive nonconformity score to use for IQT. For quaternion rotation, we require designing a score that operates in angular space. In that case, the representation of nonconformity would be over the union of position and rotation, so when you conformalize, your nonconformity score is operating on two different distance metrics. In practice, the suitable way to approach these different distance metrics is to run multiple IQT processes: one on positional uncertainty, and one on rotational. Due to this, we chose to calibrate only the xyz, the portions of the state that share the same representation to remove the confound of multiple IQT processes running. We hope to examine in future work multiple IQT processes handling these different nonconformity scores.
>
> - Then, they should be conformalized independently. To make our hardware experiments as straightforward/simple
>
> - We have updated the EnsembleDAgger hardware implementation to include the safety classifier. Our initial motivation for omitting the safety classifier on the hardware experiments was to isolate only the uncertainty quantification method, and evaluate how uncertainty-gated queries affect both approaches. In our rebuttal, we added in the safety classifier in the EnsembleDAgger implementation on hardware. See response to Reviewer YA6C for further details.

---

> > ### Comment · Reviewer_jhY9 · 2024-11-23
> > **Response**
> >
> > Thank you for your very detailed responses! I greatly appreciate the additional experiments and clarifications provided. Regarding only applying the method to end effector position, I would recommend that this limitation is included explicitly in the manuscript somewhere.

---

> > > ### Author Response · Authors · 2024-11-24
> > > **Response to Reviewer jhY922**
> > >
> > > Thank you for your response and feedback! We have updated the main text of the manuscript to clarify the limitation regarding the choice of end-effector position (line 478, highlighted in **blue**).

---

### Official Review · Reviewer_YNNu · 2024-11-03

**Soundness:** 4
**Presentation:** 4
**Contribution:** 3
**Rating:** 8
**Confidence:** 4

**Summary:**

The authors propose ConformalDAgger that uses runtime prediction interval to actively query expert’s feedback in the context of imitation learning for robot manipulation. ConformalDAgger introduces Intermittent Quantile Tracking, an uncertainty quantification method that provably minimizes regret about error of following expert’s behavior based on Angelopoulos et al., 2024, but also extends it to situations where the feedback is intermittent. Robot asks for human feedback whenever the prediction interval size exceeds some defined threshold, and the threshold also affects the regret guarantee. ConformalDAgger is validated on both typical time-series prediction settings and also robot imitation learning with human correction, showing improved coverage of expert behavior when the expert behavior distribution shifts, compared to EnsembleDAgger, a Bayesian uncertainty quantification method that can be miscalibrated.

**Strengths:**

The paper addresses a very important and timely problem in training robot manipulation policies — simple behavior cloning suffers from covariate shift, and rigorously quantifying the policy’s uncertainty and using it to query additional feedback from humans are an intuitive and potentially effective way to address covariate shift. Proper UQ ensures strong coverage while minimizing the amount of feedback needed.

ConformalDAgger is based on the online conformal prediction theory, which provides rigorous guarantees for the policy error, albeit being a regret-based one. The proposed extension to intermittent feedback setting is very novel.

ConformalDAgger shows clear advantage in coverage compared to EnsembleDAgger, which can be miscalibrated (although I have concerns about the setup of the baseline below). I also appreciate the experiments in the appendix varying p_t.

The paper is very well-written. Most experimental details are presented in the main text. All the illustrations are nicely done and convey the central ideas and experimental results cleanly.

**Weaknesses:**

I have mainly two concerns about the paper and proposed approach. The first one is about the experiment setup comparing ConformalDAgger to EnsembleDAgger. There is no discussion about how some of the hyperparameters, e.g., uncertainty threshold and safety classifier threshold used in EnsembleDAgger, are chosen in practice. I understand that these numbers can be difficult to choose in a rigorous manner, but some discussion will be greatly appreciated. Also, to strengthen the overall results, I suggest trying out a few different sets of thresholds, or even sweeping them, and comparing the two methods across the choice of these parameters. My understanding is that, with rigorous online conformal prediction theory, some of these parameters in ConformalDAgger will be easier to tune, and it would be good to discuss this too (or not if I misunderstand).

My second concern lies in the overall approach. IQT is a nice theory and provides guarantees, but in practice, if the feedback rate (p_t) is not high, which probably should be the case given human intervention can be tedious, the regret guarantee can be quite loose. The relationship between interval size and robot asking for help also confuses me: if robot doesn’t ask for help, the prediction interval is not updated since there is no new information, and intuitively this leads to less information for the robot deciding whether to ask for help. From my intuition, it makes sense to consider additional contextual information to adjust the prediction set size. I am curious how the authors think of this concern.

**Questions:**

My question for the rebuttal is about the choice of hyperparameters. Can the authors comment on the choices in Sec. 5 and Sec. 6? If it is not possible to run new experiments, what would the effect of varying the threshold and temperature on the comparison to EnsembleDAgger look like?

I also suggest adding the training details of EnsembleDAgger and e.g., how the action variance is computed, in the appendix.

---

> ### Author Response · Authors · 2024-11-22
> **Response to Reviewer YNNu: On Hyperparameter Sensitivity**
>
> We thank the reviewer for the thoughtful and constructive feedback. We are encouraged by the reviewer’s positive feedback regarding the utility of conformal prediction-based uncertainty. With respect to weaknesses and questions, we provide our responses as follows. We have uploaded a revised manuscript with additional experiments, analysis, and baselines; we discuss the results below and refer the reader to the updated figures in the manuscript. We use the color **blue** to highlight updated text and figures in the manuscript.
>
> ## A. Hyperparameter Selection
> The hyperparameter selection is indeed an important hyperparameter that affects the algorithm performance. Our approach and all baselines require thresholds for hyperparameter selection
> - For EnsembleDAgger, practitioners need to select an uncertainty threshold over the variance in ensemble predictions to flag the need for intervention, as well as a safety threshold which informs the classes (safe vs unsafe) to train the safety classifier.
> - For ConformalDAgger, we need to select only an output level uncertainty threshold.
>
>
> While there is no established rigorous method for selecting the threshold values for each algorithm, we included a new sensitivity analysis in the Expert Shift (Human-gated probability = 0.2) context. Due to the expensive training and demonstration collection time of the hardware experiments, we focused our hyperparameter sensitivity analysis in the 4D simulated reaching task.
>
> \
> &nbsp;
> ### Uncertainty threshold ablation for Conformal vs. Ensemble:
> We first varied the uncertainty threshold ($\tau$) for the two methods that maintain an uncertainty estimate: ConformalDAgger and EnsembleDAgger. We varied the uncertainty threshold across three discrete values: [0.03, 0.06, 0.09]. We note that since EnsembleDAgger relies on a safety classifier, we held the safety threshold constant throughout all uncertainty threshold ablations at 0.03. The results are averaged over 3 seeds. On one hand, we find that when $\tau$ is low (0.03), Conformal and Ensemble both identify the shift and ask for help accordingly; however, for both algorithms this comes at the cost of a great deal of human feedback requested throughout every deployment episode. Even in the first episode in which there has been no expert shift, a low-threshold at 0.03 requires conformal and ensemble to ask questions 70-80% of the time. On the other hand, when the threshold is set too high, at $\tau = 0.09$, Ensemble is unable to identify the shift and continuously mispredict. Comparatively, we find that ConformalDAgger’s performance is less sensitive to this threshold choice, and is able to effectively adapt to the expert’s shifted policy.
> \
> &nbsp;
> ### Summary:
> Comparatively, we find that ConformalDAgger’s performance is less sensitive to this threshold choice, and is able to effectively adapt to the expert’s shifted policy. This is much in line with the reviewer’s intuition that with rigorous online conformal prediction, the ConformalDAgger parameters are easier to tune, and more flexible in the choice of uncertainty threshold. We also compared hyperparameter sensitivity for our additional baselines (for these, see response to Reviewer YA6C).
>
> Please refer to Appendix Section on Hyperparameter Sensitivity Analysis (Appendix Section B) for these figures and results.

---

> ### Author Response · Authors · 2024-11-22
> **Regarding the frequency of human feedback**
>
> ## B. On Needing Sufficiently Frequent Human Feedback
>
> In practice, it is true that if the feedback rate ($p_t$) is not high, the finite horizon (T) regret guarantee can be quite loose, as reflected in Proposition 1, which relies on the value of $p_t$. Intuitively, this makes sense, as if the model does not receive enough access to the ground truth labels, it will be challenging to ensure coverage, as the intervals must be adapted based on whether the model is correctly predicting the ground truth values.
>
>
> In ConformalDAgger, the robot policy relies on having some, intermittent access to the ground truth values. In our simulated and hardware experiments, we control the level of human-gated feedback to ensure that we received a minimum of human feedback at a likelihood of 0.2. If the robot almost never receives expert feedback, the intervals may not be updated frequently enough, and may remain uncalibrated. In that context, leveraging ensemble disagreement and other sources of uncertainty in the policy can be combined with IQT to provide multiple avenues for uncertainty-prompted feedback. IQT can be combined with these other approaches to offer online, adaptive uncertainty quantification, in addition to the other mechanisms used in for EnsembleDAgger.

---

> ### Author Response · Authors · 2024-11-22
> **Response to Questions**
>
> We added the hyperparameter sensitivity analysis in Section G of the updated Appendix.
>
> We have added to the main text of the manuscript the hardware training details of EnsembleDAgger, including how action variance is computed over the position and rotation components of the policy’s prediction. The ensemble’s uncertainty threshold is 0.005, at approximately the mean of the distribution of ensembles variances (over position and orientation predicted by the policy) over the training data.

---

> > ### Comment · Reviewer_YNNu · 2024-11-24
> > **Reviewer response**
> >
> > I thank the authors for answering my concerns and questions with the newly added experiments and discussions.
> >
> > For my second concern, I think the intuition "In ConformalDAgger, the robot policy relies on having some, intermittent access to the ground truth values." makes sense and is a reasonable requirement. I don't consider it a significant weakness of the proposed approach.
> >
> > For my first concern, I appreciate you trying out different values of $\tau$, but I imagine ConformalDAgger and EnsembleDAgger may switch from ensuring coverage to miscoverage at different threshold levels (maybe with higher $\tau$ than 0.09, ConformalDAgger will miscover too?). Then my question is, do you expect ConformalDAgger to achieve overall an better ratio of coverage to help than EnsembleDAgger across a larger range of thresholds? You don't have to run new experiments but I am curious if there is a fundamental intuition based on the approaches themselves.
> >
> > Another suggestion: Could you add the condition for triggering robot query (with $\tau$) in Algorithm 1?

---

> > > ### Author Response · Authors · 2024-11-24
> > > **Response to Reviewer YNNu23: Regarding Miscoverage sensitivity**
> > >
> > > Thank you for your response and feedback! We provide our response below and changes to the manuscript have been made, uploaded, and are highlighted in **blue**.
> > > \
> > > &nbsp;
> > > ## On Miscoverage sensitivity as a function of thresholds:
> > >
> > > We divide our answer based on theoretic and empirical findings.
> > >
> > > - **Empirically:** We find that if $\tau$ is set too high (at 0.18, 0.21), ConformalDAgger may not flag high uncertainty, relying instead on human-gated interventions to update IQT. When we ablate the values of tau between the values [0.03, 0.06, 0.09, 0.12, 0.15], we found ConformalDAgger to be fairly robust in terms of miscoverage (see updated Figure 7 in Appendix B, $\tau$=0.18, 0.21 included as well).
> > >
> > > - **Theoretically:**
> > >     * If $\tau$ is set too high, failing to properly reflects the distribution of encountered misprediction residuals, ConformalDAgger will not flag high uncertainty, relying on human-gated interventions only ($P(obs^h_t|o_t)$) to update IQT and provide retraining data, which may result in poor coverage. However, importantly, one benefit of IQT is that its theoretical guarantees afford us an understanding of what the miscoverage gap will be.
> > >     * Recall that Proposition 1 gives us a bound on the miscoverage gap between the achieved miscoverage rate and desired miscoverage rate $\alpha$. This gap depends on 1) the time horizon $T$ that we are running our algorithm, and 2) the maximum value of $\frac{\gamma_t}{p_t}$.
> > >     * In cases where $\tau$ is set too high such that ($P(obs^r_t|o_t)$ is close to 0), meaning the robot almost never queries for additional help, the probability of getting feedback $p_t$ is approximately the $P(obs^h_t|o_t)$, relying only on the likelihood of human feedback. For short sequences ($T$ small), and infrequent human-gated feedback ($p_t$ small), Prop 1 informs us that this miscoverage gap might be large. Prop 1 shows the theoretical basis for that when human-gated feedback is infrequent, and highlights that a proper and effective selection of $\tau$ to raise $P(obs^r_t|o_t)$ and subsequently $p_t$, helps to reduce the miscoverage gap.
> > >
> > > &nbsp;
> > > ## On Algorithm 1 Update:
> > > Thank you for the suggestion! In Algorithm 1, we updated $P(obs_t^r|o_t ; \tau)$ (originally $P(obs_t^r|o_t)$) to more clearly denote that the triggering of the robot queries relies specifically on $\tau$.

---

> > > > ### Comment · Reviewer_YNNu · 2024-11-24
> > > > **Reviewer response**
> > > >
> > > > I see, thanks for the detailed clarifications. I have trouble opening the new revised appendix (maybe some issue with openreview itself), but I suggest adding the new discussion to the appendix if it is not added yet.
> > > >
> > > > For the algorithm, I actually meant writing out explicitly the condition triggering robot query (i.e., one in the Leveraging Uncertainty: Asking for Help paragraph) that also depends on $\beta$ and $u, and also the human intervention. Right now Line 7 still reads vague and is not self-contained.
> > > >
> > > > I have raised my score to 8 and the soundness score to 4.

---

> ### Author Response · Authors · 2024-11-24
>
> Thank you for your feedback and revised score.
>
> We have included the discussion in the Appendix Section B, and have uploaded the updated Appendix with the discussion included, with changes highlighted in blue.
>
> We added a line in Algorithm 1, which explicitly writes the robot query triggering condition (in newly uploaded manuscript main text). The additional Line 7 in the Algorithm now reads "Compute robot query likelihood (e.g. $P(\text{obs}^r_t\mid o_t;\tau) = 1$ if $u(x_t; \pi^r) > \tau$, else 0.)" to indicate that the queries are triggered when the uncertainty is greater than $\tau$, and this threshold may for example, be a hard threshold, or soft (the soft threshold defined in the Leveraging Uncertainty: Asking for Help paragraph).

---

### Official Review · Reviewer_YA6C · 2024-11-04

**Soundness:** 2
**Presentation:** 3
**Contribution:** 2
**Rating:** 6
**Confidence:** 3

**Summary:**

Interactive imitation learning with queryable experts (e.g. Dagger) are a powerful approach to robot learning from demonstration. Unlike supervised-based offline approaches like behavioral cloning, interactive IL can avoid the distribution shift between the training distribution (i.e. the expert’s state distribution) and the test-time distribution (i.e. the learner’s state distribution). Prior work in uncertainty quantification-based IL has developed methods of actively seeking feedback from the expert (robot-gated), in contrast to the original Dagger-based approach, where the expert dictated the intervention (human-gated). The paper proposes leveraging online conformal prediction as a method for uncertainty quantification during deployment time. The paper extends quantile tracking to the intermittent feedback setting, which is what is encountered in the interactive IL setting the paper considers for robot learning. Using the proposed intermittent quantile tracking (IQT), the paper proposes a new approach to uncertainty quantification-based IL that uses IQT’s prediction intervals and evaluates on simulation and physical robot tasks.

**Strengths:**

The paper motivates the problem well and presents an improvement to existing uncertainty-aware interactive IL methods (specifically Dagger-based) that incorporates uncertainty estimates in deployment time. The figures are well-done, and the experimental setup is mostly understandable (see points below).

**Weaknesses:**

Overall, the limitations in the paper’s experimental results are the most significant weaknesses of the paper, which I discuss in detail below.

The paper only compares to one baseline, and they do not provide justification for why other baselines are not considered. Either incorporating additional baselines or providing a discussion of why other Dagger-based or Monte Carlo-based (e.g. Cui et al. 2019) methods were not considered and would strengthen the empirical results.

Continuing with the limitations of the baselines, in the Hardware Experiments section (Sec. 6), the paper implements EnsembleDagger without a safety classifier. Rather than train a safety classifier to detect dissimilarity between the expert’s actions and the robot’s actions, the diffusion policy is sampled 3 times in order to “capture the notion of epistemic uncertainty.” The justification for this decision is unclear. Is replacing the safety constraint with multiple samples from the diffusion policy (which will be stochastic) better than implementing a safety classifier? Or is implementing a safety classifier unfeasible? (Additionally, I was unable to find the relevant discussion of an *implicit* EnsembleDagger-type approach in the referenced Wolleb et al. 2022.)

There are no details of the parameter tuning that was done to select EnsembleDagger’s uncertainty threshold. This is quite an important parameter that can affect its performance. EnsembleDagger is the only baseline compared against, so the lack of discussion of the uncertainty parameter undermines the relevance of the baseline.

More broadly, the empirical results only consider distribution shift due to expert shift. The paper does not test on environments with environment shift (e.g. changing dynamics).

On a different but important dimension, Dagger is also known to be a resource-intensive training procedure, requiring a queryable expert. The paper currently lacks a discussion of the computational efficiency of ConformalDagger and how it compares to EnsembleDagger. (e.g. measuring the wall-clock time to establish a comparison, discussing the IQT procedure’s efficiency when integrated into Algorithm 1, etc).

As a further extension, the original Dagger is known to fail in the setting where the expert policy is unrealizable. Currently, the paper has no discussion on what setting they consider with respect to the expert policy and the learner’s policy class, and whether the ConformalDagger approach would fare better or worse than prior work in the misspecified setting.

**Minor Comments:**
- Section 3 would benefit from further explanation (e.g. explaining the significance of Proposition 1) would improve the clarity of the section.
- Appendix section numbers are listed as ???.

**Questions:**

The phrasing of Line 7 in Algorithm 1 is confusing, as it suggests two of the exact same samplings.

---

> ### Author Response · Authors · 2024-11-22
> **Response to Reviewer YA6C: Regarding Expert Shift Context**
>
> We thank the reviewer for the thoughtful and constructive feedback. With respect to weaknesses and questions, we provide our responses as follows. We have uploaded a revised manuscript with additional experiments, analysis, and baselines; we discuss the results below and refer the reader to the updated figures in the manuscript. We use the color **blue** to highlight updated text and figures in the manuscript.
> \
> &nbsp;
> ## A. Regarding expert shift and environment shift.
> We would first like to clarify – algorithmically, our IQT approach is agnostic to the underlying cause of distribution shift between the expert and the policy. All that it requires is accessing the expert’s labels intermittently to construct calibrated intervals on the base policy’s predictions. The reason the base policy is inconsistent with the expert labels can come from a variety of underlying causal factors. In our initial submission, we focused on expert shift – wherein the expert’s latent strategy changes over times – as one concrete cause of why the base policy disagrees with the expert.
>
> More broadly, policy mispredictions during deployment may result from a multitude of reasons, including suboptimal or multimodal demonstrations, covariate shift (e.g., encountering new observations of the environment during policy execution), or expert shift. To show the generality of our IQT algorithm, we added a **Environment Shift** context, where the state distribution during deployment differs from training. We instantiated **Environment Shift** in both the toy 4D reaching task and the newly-added **Robosuite benchmark Nut-Assembly task** (Robosuite benchmark results can be found in Appendix Section G in revised manuscript). Results for the Simulated 4D Reaching Task are detailed in rebuttal point “B” below and in the revised manuscript (Figure 5 in Section 5.2 in revised manuscript main text).

---

> ### Author Response · Authors · 2024-11-22
> **Inclusion of Additional Baselines**
>
> ## B. Regarding additional baselines
> We appreciate the suggestion of including additional interactive imitation learning baselines! In our original manuscript, we focused on comparing ConformalDAgger with EnsembleDAgger because the ensemble output variance provides a measurable interval for coverage, enabling direct comparison. As suggested by the reviewer, we included two relevant DAgger approaches: **LazyDAgger**[2] and **SafeDAgger** [3]. SafeDAgger relies on switching to human control when the predicted action discrepancy exceeds a threshold, while LazyDAgger initiates human intervention based on a safety classifier and returns to autonomous control only when the discrepancy falls below an additional context-switching-threshold [2]. Similar to Ensembles, Monte-Carlo Dropout approaches (e.g. UAIL [1]) can also be used to estimate epistemic uncertainty in deep networks; we chose EnsembleDAgger because of prior works empirically observed by Lakshminarayanan et al. [2017], Lee et al. [2015] that training ensemble networks with random initialization works well in practice for capturing this epistemic uncertainty.
>
> We have rerun experiments including these additional baselines and under broader environment shift conditions. The results are discussed in the section below and figures are in the attached PDF (Figure 5 in Section 5.2 in revised manuscript).
> \
> &nbsp;
> ### Updated Results on Simulated 4DoF Reaching Task with Human Gated Probability (=0.2 and =0.5):
> When the expert shifts and drifts, ConformalDAgger asks for more help immediately compared to EnsembleDAgger and SafeDAgger, enabling the algorithm to more quickly align to the expert, with fewer human interventions than LazyDAgger. LazyDAgger does successful align to the shifted strategy, but requests many more interventions overall.
>
> Under environment shift, EnsembleDAgger is well-calibrated, boosting the number of interventions requested. This is intuitive as ensembles capture the variance in the training distribution and when the environment shifts, changing the training distribution, the algorithm flags high uncertainty. ConformalDAgger performs well on environment shift context, though asks fewer questions than EnsembleDAgger, converging to the shifted policy nearly as quickly as EnsembleDAgger.
>
> When the human-gated probability is high (0.9), as expected, all approaches receive high quantities of human feedback, resulting in rapid converge to expert and environment shift.
>
> [1] Y. Cui, D. Isele, S. Niekum and K. Fujimura, "Uncertainty-Aware Data Aggregation for Deep Imitation Learning," 2019 International Conference on Robotics and Automation (ICRA), Montreal, QC, Canada, 2019, pp. 761-767, doi: 10.1109/ICRA.2019.8794025.
>
> [2] Hoque, Ryan, et al. "Lazydagger: Reducing context switching in interactive imitation learning." 2021 IEEE 17th international conference on automation science and engineering (case). IEEE, 2021.
>
> [3] Zhang, Jiakai, and Kyunghyun Cho. "Query-efficient imitation learning for end-to-end autonomous driving." arXiv preprint arXiv:1605.06450 (2016).

---

> ### Author Response · Authors · 2024-11-22
> **Regarding Hardware Experiments - EnsembleDAgger Safety Classifier**
>
> ## C. Hardware Experiments - EnsembleDAgger Safety Classifier
>
> We have updated the EnsembleDAgger hardware implementation to include the safety classifier. Our initial motivation for omitting the safety classifier on the hardware experiments was to isolate only the uncertainty quantification method, and evaluate how uncertainty-gated queries affect both approaches. In our rebuttal, we added in the safety classifier in the EnsembleDAgger implementation on hardware.
>
> &nbsp;
> ### Safety Classifier Training Details:
> We trained the safety classifier as a fully connected network which takes in as inputs the Resnet-18 embeddings of the camera images and robot proprioceptive state. We will include training details in the manuscript. For the EnsembleDAgger threshold, we chose our threshold of 0.005 at the mean of the distribution of ensembles variances over the training data. For the safety classifier, we found that setting the safety threshold equal to the mean divergences between expert and learner policy was effective in achieving sufficient safety-prompted queries. Wolleb et al. 2022 takes an implicit-Ensemble by sampling multiple times from the same diffusion model, but its domain is not in the content of interactive imitation learning. We only use the paper’s notion of extracting an ensemble from multiple samples from the same diffusion model to produce our ensemble. We will clarify this in the revised manuscript.
>
> &nbsp;
> ### Updated Hardware Experiment Details:
> For the updated hardware experiments in the rebuttal, we trained on 50 demos for the base policy, and perfomed an interactive deployment with all methods (Conformal, Lazy, Safe, and Ensemble) on 24 interactive deployments. We used a buffer size of 6k rather than 12k, cutting it in half.
>
> &nbsp;
> ### Updated Hardware Results:
> We find that without shift, all approaches refrain from robot-gated human interventions. With expert shift, ConformalDAgger learns the shifted policy most saliently. When Lazydagger asks for help, it’s intervention profile is similar to ConformalDAgger, but because the algorithm doesn’t ask for consistently in each episode, it does not receive enough additional data to sufficiently adapt to the shifted policy. Figures are updated to reflect these results in the revised manuscript (Figure 6 in Section 6).

---

> ### Author Response · Authors · 2024-11-22
> **Regarding Hyperparameter Sensitivity**
>
> ## D. Ensemble Hyperparameter Selection
> The hyperparameter selection is indeed an important hyperparameter that affects the algorithm performance.
> ConformalDAgger, EnsembleDAgger, LazyDAgger, and SafeDAgger all rely on a choice of threshold.
> - For EnsembleDAgger, practitioners need to select an uncertainty threshold over the variance in ensemble predictions to flag the need for intervention, as well as a safety threshold which informs the classes (safe vs unsafe) to train the safety classifier. The safety threshold is needed for SafeDAgger.
> - For ConformalDAgger, we need to select only an output level uncertainty threshold.
> - For LazyDAgger, the safety threshold is needed, as well as a context switching threshold.
>
> While there is no established rigorous method for selecting the threshold values for each algorithm, we included a new sensitivity analysis in the Expert Shift (Human-gated probability = 0.2) context.
>
>
> &nbsp;
> ### Uncertainty threshold ablation for Conformal vs. Ensemble:
> We first varied the uncertainty threshold ($\tau$) for the two methods that maintain an uncertainty estimate: ConformalDAgger and EnsembleDAgger. We varied the uncertainty threshold across three discrete values: [0.03, 0.06, 0.09]. We note that since EnsembleDAgger relies on a safety classifier, we held the safety threshold constant throughout all uncertainty threshold ablations at 0.03. The results are averaged over 3 seeds. On one hand, we find that when $\tau$ is low (0.03), Conformal and Ensemble both identify the shift and ask for help accordingly; however, for both algorithms this comes at the cost of a great deal of human feedback requested throughout every deployment episode. Even in the first episode in which there has been no expert shift, a low-threshold at 0.03 requires conformal and ensemble to ask questions 70-80% of the time. On the other hand, when the threshold is set too high, at $\tau = 0.09$, Ensemble is unable to identify the shift and continuously mispredict. Comparatively, we find that ConformalDAgger’s performance is less sensitive to this threshold choice, and is able to effectively adapt to the expert’s shifted policy.
>
>
> &nbsp;
> ### Context-Switching threshold ablation for LazyDAgger:
> LazyDAgger relies on a context-switching threshold, which we ablated with values of 0.1, 1, and 10, as in done in the LazyDagger paper [cite]. Similarly to the prior experiments, we hold the LazyDAgger safety threshold constant at 0.03. These values are multipliers on the safety threshold used to train the classifier and inform switches back to the robot after the safety classifier as flagged an unsafe state. We find that thresholds of 1 and 10 results in highly infrequent questions during the episodes where no shift has occurred, but that 0.1 sufficiently better balances the queries.
>
>
> &nbsp;
> ### Safety threshold ablation for SafeDAgger:
> SafeDAgger relies on a safety classifier threshold,  which we ablated with values 0.01, 0.03, and 0.05. Because SafeDAgger differs from Ensemble and Lazy in that it only has one mechanism for inducing queries, we find that 0.03 and 0.05 are insufficient are generating queries when the safety classifier is used in isolation, and 0.01 sufficiently better balances the queries.
>
> \
> &nbsp;
> ### Summary:
> All results are fully reflected in the revised manuscript. Please see Appendix Section on Hyperparameter Sensitivity Analysis (Appendix Section B) for these figures and results.

---

> ### Author Response · Authors · 2024-11-22
> **Regarding Expert Realizability, Algorithm Runtimes, and Minor Comments/Questions**
>
> ## E. Regarding Expert Realizability
> We agree with the reviewer that an outstanding challenge in the theory of DAgger is guarunteeing learning a high-quality policy from the expert when the expert is unrealizable. We assume, much like [1], that our expert is realizable. Whether or not this is true in practice with human experts is an open area of research. Nevertheless, similar to HG-DAgger[3] and EnsembleDAgger[3], our approach offers a practical approach to contend with these real-world challenges. We included this discussion in the manuscript.
>
> [1] Spencer, Jonathan, et al. "Feedback in imitation learning: The three regimes of covariate shift." arXiv preprint arXiv:2102.02872 (2021).
>
> [2] Menda, Kunal, Katherine Driggs-Campbell, and Mykel J. Kochenderfer. "Ensembledagger: A bayesian approach to safe imitation learning." 2019 IEEE/RSJ International Conference on Intelligent Robots and Systems (IROS). IEEE, 2019.
>
> [3] Kelly, Michael, et al. "Hg-dagger: Interactive imitation learning with human experts." 2019 International Conference on Robotics and Automation (ICRA). IEEE, 2019.
>
>
> &nbsp;
> ## F. Regarding Algorithm Runtimes
> Quantifying uncertainty via IQT during policy rollout on an Apple M3 Max chip takes on average 0.0013 ms per timestep during interactive rollouts of the simulated 4DoF reaching task experiments.
>
> &nbsp;
> ## Response to Minor Comments and Questions:
>
> We included further clarification about the significance of Proposition 1 in Section 3. As included in the manuscript, we describe that Prop 1 guarantees IQT gives the $1-\alpha$ long-term empirical coverage frequency, even with intermittent ground truth access, regardless of the underlying data generation process.
>
> We clarified line 7 of Algorithm 1 in the manuscript. The robot elects to ask questions with probability $P(obs^r_t|o_t)$. Alternatively, we clarify that the human may also intervene independently. The human intervenes independently with probability $P(obs^h_t|o_t)$. These samplings are drawn from two different probability distributions: the likelihood of asking for help, and the likelihood of the human independently giving feedback irrespective of the robot’s query for help.
>
> The Appendix numbers have been fixed.

---

> > ### Comment · Reviewer_YA6C · 2024-12-02
> > **Response**
> >
> > Thank you for your thorough response. Regarding the expert realizability assumption, I believe this should be stated clearly as an assumption in the main section of the paper. (It may also be worth acknowledging the related work that imposes the assumption as well.) I have raised my score accordingly.

---

> > > ### Author Response · Authors · 2024-12-02
> > >
> > > Thank you for your feedback and revised score. We will ensure to state clearly the expert realizability assumption and discussion in the main section of the paper.

---

### Author Response · Authors · 2024-11-27
**Overall feedback and rebuttal summarization**

We sincerely thank the reviewers for their time and constructive feedback, which we greatly appreciate. We are particularly encouraged by the reviewers’ recognition of the paper’s strengths, including its technical novelty and empirical effectiveness.

 We prepared detailed rebuttal for reviewers' concerns on paper weaknesses. We are happy that all reviewers (jhY9, YNNu, and YA6C) acknowledged our responses, with Reviewers YNNu and YA6C kindly raising the score. Based on these discussions, we have revised the paper accordingly. Detailed revisions, which are marked **blue** in the updated pdf file, include:

- A. **Discussing Generality of Expert Shift and Inclusion of Environment Shift Context**
To show the generality of our IQT algorithm, we added a **Environment Shift** context, where the state distribution during deployment differs from training. We instantiated **Environment Shift** in both the toy 4D reaching task (Section 5) and the newly-added **Robosuite benchmark Nut-Assembly task** (Robosuite benchmark results can be found in Appendix Section G in revised manuscript).

- B. **Inclusion of Additional Baselines**
Following the reviewers’ suggestions, we included two additional DAgger baselines: LazyDAgger [2] and SafeDAgger [3]. We reran experiments to include these baselines under broader environment shift conditions, and the results are presented in Figure 5 (Section 5.2) and Appendix Section G of the revised manuscript.

- C. **Hardware Implementation Safety Classifier**
We updated the EnsembleDAgger hardware implementation to include a safety classifier and reported hardware results with the additional baselines (Section 6).

- D. **Hyperparameter Sensitivity Analysis**
In Appendix Section B, we included a sensitivity analysis for threshold hyperparameter selection for ConformalDAgger and baseline methods.

As the author-reviewer discussion period concludes, we remain available to address any outstanding questions or concerns and are open to further revisions. We once again thank the reviewers for their thoughtful feedback and careful evaluation of our submission.

---

### Meta-Review · Area_Chair_5A6F · 2024-12-17

**Metareview:**

The paper presents ConformalDAgger, which combines online conformal prediction with interactive imitation learning. The key contributions include:
- A novel algorithm called Intermittent Quantile Tracking (IQT) that adapts conformal prediction to handle intermittent expert feedback, with theoretical coverage guarantees
- Integration of IQT with DAgger to create ConformalDAgger, enabling uncertainty-based expert querying during deployment
- Evaluation on both synthetic and real robotic tasks showing the method's effectiveness in handling expert policy shifts

Weaknesses:
- Initial submission compared against only one baseline (EnsembleDAgger), though this was addressed in revision with additional baselines
- Limited exploration of hyperparameter sensitivity (addressed in revision)
- Current implementation only applies IQT to end-effector position, not orientation
- Real robot experiments focused mainly on a single task type

**Additional Comments On Reviewer Discussion:**

Initial Main Comments:
- Reviewer jhY9 questioned the practical relevance of expert shift scenarios and suggested additional baselines and simulated benchmarks
- Reviewer YNNu raised concerns about hyperparameter sensitivity and implementation details
- Reviewer YA6C requested clarification of baseline implementation details and additional experimental validation

Key changes from author responses include:
- Addition of LazyDAgger and SafeDAgger as baselines
- New experiments on the Robosuite benchmark
- Comprehensive hyperparameter sensitivity analysis
- Clarification of EnsembleDAgger implementation details
- Additional discussion of applicability beyond expert shift

The authors provided thorough responses and made substantial revisions, leading reviewers to express satisfaction with the changes. Reviewer YNNu explicitly upgraded their score to 8 based on the responses.

---

### Decision · Program_Chairs · 2025-01-22

Accept (Poster)